# Ancient proteins provide evidence of dairy consumption in eastern Africa

Madeleine Bleasdale [1,2✉], Kristine K. Richter[1], Anneke Janzen[1,3], Samantha Brown [1], Ashley Scott [4], Jana Zech[1], Shevan Wilkin [1], Ke Wang [4], Stephan Schiffels [4], Jocelyne Desideri [5], Marie Besse [5], Jacques Reinold[6], Mohamed Saad[7], Hiba Babiker [8], Robert C. Power [1,9], Emmanuel Ndiema[1,10], Christine Ogola[10], Fredrick K. Manthi[10], Muhammad Zahir [1,11], Michael Petraglia [1,12,13], Christian Trachsel[14], Paolo Nanni [14], Jonas Grossmann [14], Jessica Hendy [1,15], Alison Crowther[1,12], Patrick Roberts [1,12], Steven T. Goldstein [1] & Nicole Boivin [1,12,13,16✉]

Consuming the milk of other species is a unique adaptation of *Homo sapiens*, with implications for health, birth spacing and evolution. Key questions nonetheless remain regarding the origins of dairying and its relationship to the genetically-determined ability to drink milk into adulthood through lactase persistence (LP). As a major centre of LP diversity, Africa is of significant interest to the evolution of dairying. Here we report proteomic evidence for milk consumption in ancient Africa. Using liquid chromatography tandem mass spectrometry (LC-MS/MS) we identify dairy proteins in human dental calculus from northeastern Africa, directly demonstrating milk consumption at least six millennia ago. Our findings indicate that pastoralist groups were drinking milk as soon as herding spread into eastern Africa, at a time when the genetic adaptation for milk digestion was absent or rare. Our study links LP status in specific ancient individuals with direct evidence for their consumption of dairy products.

[1] Department of Archaeology, Max Planck Institute for the Science of Human History, Jena, Germany. [2] Department of Archaeology, University of York, King's Manor, Exhibition Square, York YO1 7EP, UK. [3] Department of Anthropology, University of Tennessee, Knoxville, TN, USA. [4] Department of Archaeogenetics, Max Planck Institute for the Science of Human History, Jena, Germany. [5] Laboratory of Prehistoric Archaeology and Anthropology, Department F.-A. Forel for Environmental and Aquatic Sciences, Université de Genève, Geneva, Switzerland. [6] Section française de la Direction des antiquités du Soudan, Khartoum, Sudan. [7] National Corporation for Antiquities and Museums of Sudan, M.Bolheim Bioarchaeology Laboratory, Khartoum, Sudan. [8] Department of Linguistic and Cultural Evolution, Max Planck Institute for the Science of Human History, Jena, Germany. [9] Institute for Pre-and Protohistoric Archaeology and Archaeology of the Roman Provinces, Ludwig-Maximilians-University Munich, Munich, Germany. [10] Department of Earth Sciences, National Museums of Kenya, Nairobi, Kenya. [11] Department of Archaeology, Hazara University, Mansehra, Pakistan. [12] School of Social Science, The University of Queensland, Brisbane, QLD, Australia. [13] Department of Anthropology, National Museum of Natural History, Smithsonian Institution, Washington, DA, USA. [14] Functional Genomics Center, University of Zurich/ETH, Zurich, Switzerland. [15] BioArCh, Department of Archaeology, University of York, York, UK. [16] Department of Anthropology and Archaeology, University of Calgary, Calgary, AB, Canada. ✉email: bleasdale@shh.mpg.de; boivin@shh.mpg.de

Human experimentation with collecting, processing, and consuming milk from other animals enabled one of the most profound revolutions in human diet since the initial emergence of agriculture. Animal milk is rich in proteins, fat, and micronutrients and, particularly in arid environments, provides an important way of converting scarce natural resources into a portable, renewable food source[1]. Animal milk also offers the opportunity for early weaning, and can play a role in reducing birth spacing, with significant demographic implications[2,3]. The transition to consuming animal milk is thought to have been so important in human history that it drove intensive selection for lactase persistence (LP), a genetic adaptation among certain populations across Asia, Europe, and Africa that allows people to digest lactose into adulthood[4–6]. The strong selection of LP-conferring alleles in the context of milk consumption represents one of the most widely cited examples of gene-culture coevolution[5,7,8].

In spite of its status as a textbook example of gene-culture coevolution[7], however, many questions remain about the emergence of LP. In particular, the question of whether milk drinking drove the selection for LP, or if low frequencies of LP encouraged milk drinking, has not been adequately answered[5,9]. It is still unclear what selective pressures drove LP to such high frequencies, since numerous populations manage to consume milk in the absence of this genetic adaptation, possibly through external fermentation (yoghurt, cheeses) and/or microbiome adaptations[10–12]. Additionally, the early spread of livestock milking practices, and the global timing and pattern for LP emergence are poorly understood. Initial hypotheses that LP emerged rapidly among early farmers have been challenged by ancient DNA (aDNA) studies that demonstrate low frequencies of the LP-allele during the Neolithic period (~8000–4000 cal. BP) in Europe[13,14], and reports that present-day LP allele frequency levels in Germany were only attained in the last 850 years or so[15]. Palaeoproteomic studies have demonstrated that, in Europe, milk drinking was present thousands of years prior to the emergence of high LP frequencies[16], while in Mongolia prehistoric populations lacking any known LP alleles were regularly consuming milk products[17].

Forms of mobile animal pastoralism have supported large populations in Africa's arid and semi-arid grassland for several thousand years, with milk from cattle, sheep, and goat herds continuing to be vital food sources for millions of people in Africa today[18,19]. The importance of animal milk on the continent is probably key to understanding why Africa today hosts the greatest diversity of LP variants in the world[4,20,21]. All five of the known LP-associated variants are present in contemporary African populations, and three of these are believed to have independently emerged in northeastern or eastern parts of the continent, where hotspots of high LP frequencies are found today[4,5,21,22]. Despite this, the historical relationship between LP selection and milking in Africa is vastly under-studied. Counterintuitive patterns, such as the persistence of high LP frequencies among foraging populations in eastern Africa that are not assumed to have deep histories of animal management or milk consumption (e.g. Hadza), and low LP frequencies among pastoralist populations that regularly consume dairy products (e.g. Dinka), are not fully understood[4,5]. Furthermore, repeated ancient and historical population movements in Africa problematize attempts to assess the selective relationship between LP and environmental context (e.g., aridity, ultraviolet radiation[23]).

The growing corpus of aDNA research for Africa, which has generated genetic data for over 100 individuals[24–33] has so far identified only one LP-associated allele in a single individual from northern Tanzania (~2000 cal. BP)[32]. Establishing the true frequency of LP in ancient Africa is constrained by the relatively small ancient genomic dataset for the continent. In addition, the absence of an LP-associated allele in some individuals can be attributed to poor coverage on LP-associated genomic regions rather than a true absence of the LP-related genetic signature. Nevertheless, current available evidence places LP in Africa ~6000 years after the introduction of livestock to the continent, and ~1000 years after the development of specialized cattle, sheep, and goat herding economies in eastern Africa[34,35]. As in Europe, higher frequencies of LP may have emerged relatively late and well after the beginnings of dairy consumption in Africa, though more chronological information for early milk drinking and high-quality ancient LP data is needed to test this.

While researchers assume that milk consumption must have existed before LP-conferring alleles could be subjected to positive selection, there is uncertainty surrounding when and where economic strategies emphasizing milk production first emerged in Africa. The remains of domesticated cattle, sheep, and goats[35] and images of milking scenes in rock art[36] attest to early herding in Africa, but neither provide strong evidence for the extent or antiquity of dairying practices. In northern Africa, the identification of milk fats in ceramics together with the presence of domestic fauna suggest that early pastoralists were consuming milk by c. 7200 cal. BP in Libya[37,38] and by c. 6600 cal. BP in Sudan[38,39]. A recent lipid residue study of ceramics ($n = 40$) from early herder sites in the Lake Turkana Basin identified one sample with a $\Delta^{13}C$ value consistent with ruminant milk fats, suggesting some use of dairy products by 5000 BP in eastern Africa[40]. The lipid analysis also returned three other positive results (and one possible positive result) for milk fats from c. 3000-year-old ceramics ($n = 85$) from southern Kenya and northern Tanzania[40].

Lipid residue findings thus suggest early pastoral use of milk, but lipid analyses for milk residues suffer a number of constraints. For example, lipid evidence sometimes encounters issues of equifinality, particularly when local isotopic baselines are lacking, as mixing different food sources can mask dairy signatures[41]. A more important concern in the present case is that the identification of dairy fats in ceramic vessels does not necessarily mean milk was regularly consumed as food, as it is known to have medicinal[42] and ritual uses[43]. This concern is heightened when recovery rates are low. While the recent eastern African lipid study[40] appears to confirm early pastoralist use of milk, it remains unclear what the very low recovery rates of lipids with $\Delta^{13}C$ values within the range of milk fats mean in terms of consistency or frequency of milk consumption in the past. Ceramic lipid residue analysis is also unable to link milk consumption to particular individuals in a population that may have had mixed LP frequencies.

Palaeoproteomics provides a more direct method for detecting the consumption of milk by ancient humans. The extraction and identification of proteins from ancient dental calculus[16,17,44–46] not only enables the identification of milk drinking in specific individuals but also in some cases the animal from which the milk was derived, since some dairy peptides are genus- or species-specific due to single amino acid polymorphisms in different taxa. Typically, these studies have identified the milk whey protein β-lactoglobulin as opposed to other proteins more abundant in whole milk, such as caseins, although the reason for this detection bias is not yet well-understood[45]. Proteomic analysis of dental calculus can be combined with mortuary or aDNA information for specific individuals, allowing dietary patterns to be linked to circumstances like age, sex, status and, especially, the existence of LP alleles.

Here, we draw on this application to examine the origins of milk consumption relative to existing genetic data on LP for ancient individuals in eastern Africa. We use liquid chromatography

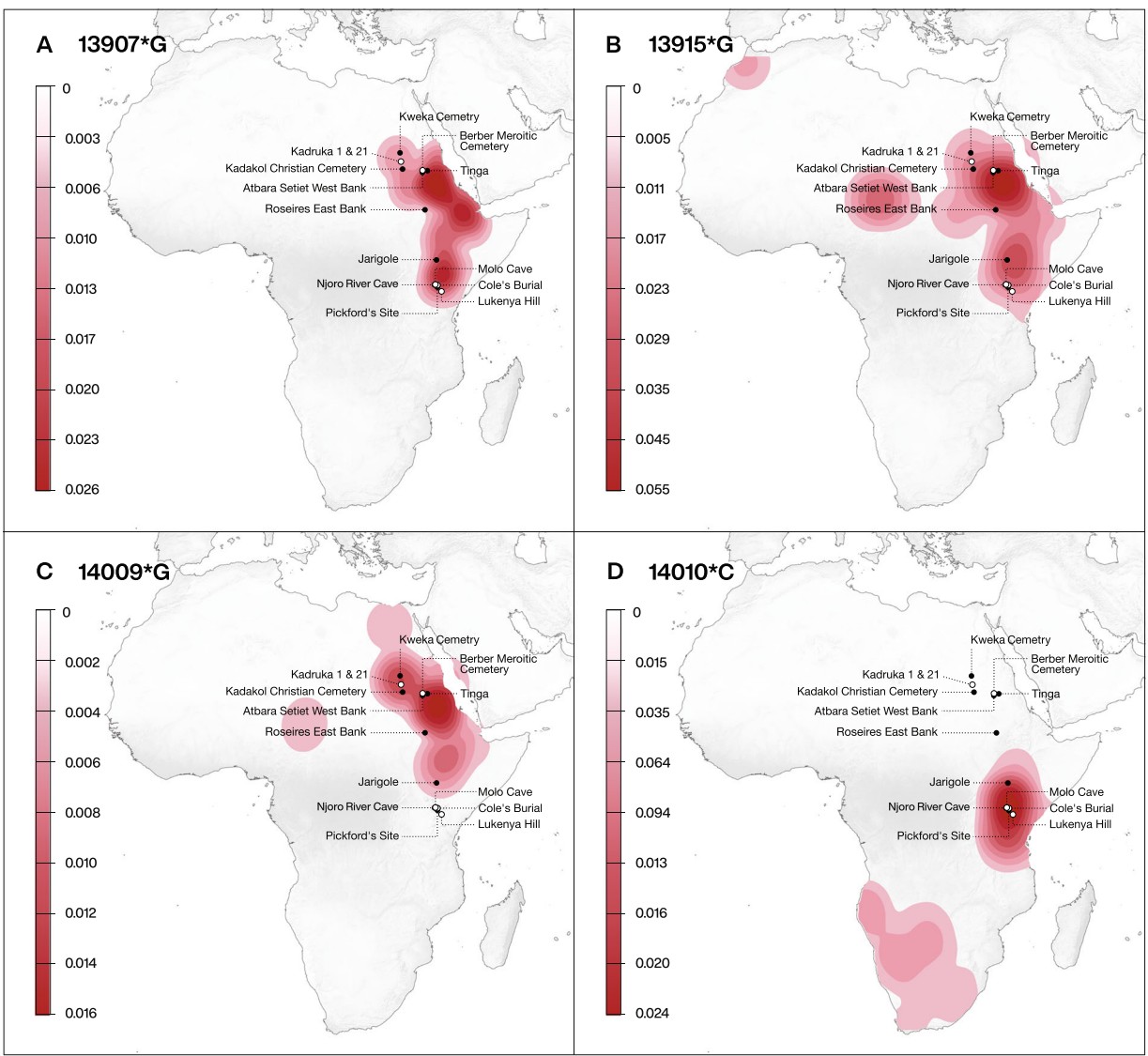

**Fig. 1 Locations of sampled archaeological sites in Kenya and Sudan in relation to the distribution of main lactase persistence alleles found in modern populations in Africa. A** 13907*G, **B** 13915*G, **C** 14009*G, and **D** 14010*C. Filled circles represent sites where milk proteins were identified in dental calculus samples, and empty circles are sites where samples did not yield milk proteins. The map was created for this study by Michelle O'Reilly (Graphic Designer for the Max Planck Institute for the Science of Human History, Jena, Germany) using QGIS 3.12 [https://qgis.org/en/site/] and the Natural Earth Database from [https://www.naturalearthdata.com/downloads/] and Adobe Illustrator CC. Heat maps were generated using published LP distribution frequencies[6].

tandem mass spectrometry (LC-MS/MS) to analyse proteins extracted from the calculus of 41 human individuals from 13 sites across Sudan and Kenya. The sites from Sudan span the Neolithic (~8000–5500 cal. BP) to Meroitic (~2300–1600 cal. BP) periods and the sites from Kenya all date to the Pastoral Neolithic (~3500–1200 cal. BP) (Supplementary Note 1, Supplementary Tables 1 and 2 and Supplementary Data 1). Today, these regions of eastern Africa display high diversity of LP-associated alleles (Fig. 1) and have yielded some of the earliest evidence for pastoralism in Africa, making them key geographical loci for exploring the spread of herding from northeastern into sub-Saharan Africa. To further assess whether these communities were reliant on animal-derived products (milk and/or meat), a total of 17 humans (including six whose dental calculus yielded milk proteins) and associated fauna from three Kenyan sites were also analysed using stable carbon ($\delta^{13}$C) nitrogen ($\delta^{15}$N) analysis of bone collagen and $\delta^{13}$C analysis of tooth enamel (Supplementary Note 4, Supplementary Data 8 and Supplementary Data 9). Stable oxygen ($\delta^{18}$O) isotope data for the

tooth enamel samples are also presented. Ancient genetic information[31,32] was obtained previously from several of the individuals tested here, as well as from other individuals from the sites sampled in the present study, enabling comparison of dairying status, ancestry, and LP allele-related data for a number of individuals and sites.

## Results
We identified milk peptides, in some cases genus- or species-specific, in the dental calculus of eight individuals, deriving from two sites in Sudan and three sites in Kenya (Fig. 2, Supplementary Tables 1 and 2 and Supplementary Data 6). The earliest milk peptides recovered are from Sudan, from a c. sixth millennium BP individual; milk was also identified as early as the fourth millennium BP in Kenya. These findings provide direct evidence of milk consumption in prehistoric Africa, and demonstrate the importance of animal milk for early pastoralists south of the Sahara.

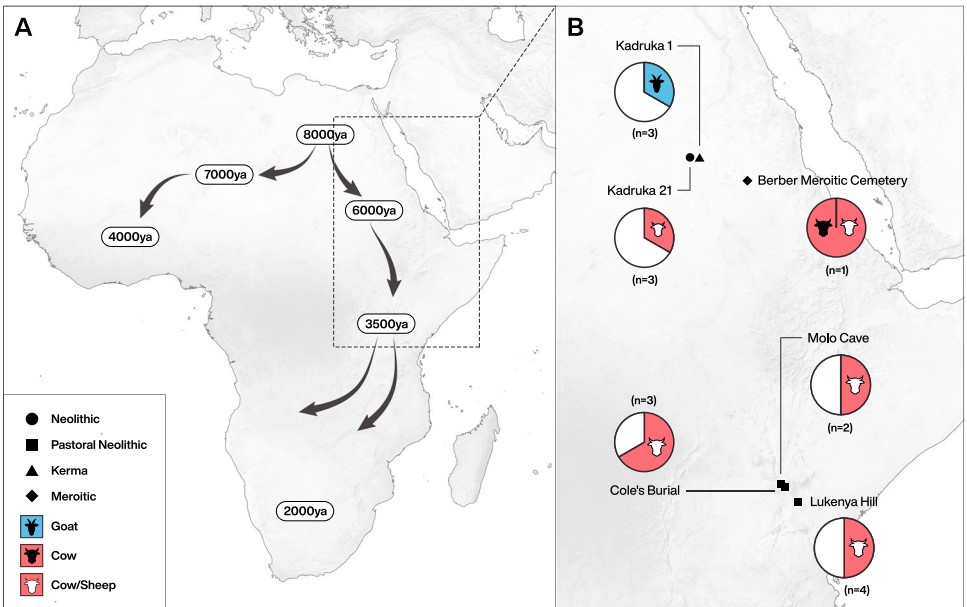

**Fig. 2 Map of sites with calculus containing milk proteins. A** Area of study in relation to the spread of cattle-based pastoralism across Africa (after Marshall and Hildebrand[35]). **B** Pie charts showing the number of individuals per site with milk proteins (shaded) proportionate to the total number of individuals that passed screening with Oral Signature Screening Database (see "Methods" and Supplementary Note 3 for full details). Neolithic: ~8000–5500 cal. BP; Kerma: ~4450–3450 cal. BP; Pastoral Neolithic: ~3500–1200 cal. BP; Meroitic: ~2300–1600 cal. BP. The maps were created for this study by Michelle O'Reilly (Graphic Designer for the Max Planck Institute for the Science of Human History, Jena, Germany) using QGIS 3.12 [https://qgis.org/en/site/] and the Natural Earth Database from [https://www.naturalearthdata.com/downloads/]. Additional edits were made using Adobe Illustrator CC.

The total number of proteins per calculus sample was variable (Supplementary Data 2) and lower than those previously reported in studies of archaeological calculus from Europe and Central Asia[16,44–47], likely reflecting poorer biomolecular preservation in the warmer environments of the African continent. Research has shown that dental calculus traps molecular signatures of the human oral proteome and oral microbiome[47,48]; therefore, well-preserved ancient dental calculus proteomes should contain an oral signature. Since some samples derived from extensively handled museum collections, each sample was searched against a custom-made, oral signature screening database (hereafter OSSD) to avoid conducting analysis for dietary proteins on samples for which oral signatures did not support authenticity (for full details of the OSSD see Supplementary Note 3). Dental calculus samples (n = 21) from 19 individuals across eight sites (46% of the total individuals analysed) passed the OSSD threshold (Supplementary Data 3).

Of the 21 samples (19 individuals) with a characteristic oral proteomic signature, 11 samples (10 individuals) had at least one milk peptide. Nine of these samples (8 individuals) were considered to have authentic dairy proteins after screening (Fig. 2, "Methods"). All nine samples had peptides from the milk whey protein β-lactoglobulin (BLG) (Supplementary Data 6 and Supplementary Figs. 1–8). The most common peptide recovered was at position 143 (Supplementary Data 6) accounting for 57% (30/52) of the total BLG peptides identified, consistent with other studies reporting milk proteins in ancient calculus[16,17,45,46]. BLG proteins from one individual from Sudan produced species-specific information, providing evidence of goat milk consumption. For all other individuals (n = 6) only genus, subfamily, family, or infraorder-level identifications could be made for the type of milk consumed (Supplementary Tables 4 and 5 and Supplementary Data 6). In addition, casein proteins, which constitute c. 80% of the proteins in cow, sheep, and goat's milk, were recovered in the calculus of one individual (Supplementary Fig. 8). No evidence

consistent with camel milk was recovered, which was not unexpected since camels were introduced from Arabia in the last c. 2000 years and only reached large numbers by ~1600 BP.

**Dairying evidence in prehistoric northeastern Africa (Sudan).** We analysed dental calculus from individuals from seven sites across Sudan, and detected milk peptides (BLG and caseins) at the sites of Kadruka and Berber Meroitic Cemetery. Dairy proteins were identified in the calculus of two individuals from Kadruka, a series of cemeteries dating from the Neolithic (~8000–5500 cal. BP) to Kerma period (~4450–3450 cal. BP) located in the Northern Dongola Reach, south of the third Cataract of the Nile[49] (Supplementary Note 1). One individual from Kadruka 21 (Z452, DA351), a Neolithic cemetery dating to ~6000 cal. BP[49] possessed milk proteins assigned to Bovinae (domestic cow/zebu) or *Ovis* (sheep). We detected BLG peptides associated with *Capra* (goat) in the calculus of an individual from Kadruka 1 (Z708) (Fig. 3), directly radiocarbon dated to 4140–3930 cal. BP (Supplementary Table 2 and Supplementary Fig. 14), providing the earliest direct evidence of goat milk consumption in Africa. Faunal remains recovered from Kadruka 1 and 21 include cattle, sheep, and goat[50,51], in line with the proteomic results. BLG peptides derived from Bovinae or *Ovis* were also present in the calculus of an individual (DA156) from the Berber Meroitic Cemetery site (~2300–1600 cal. BP) in northern Sudan. In the calculus of this individual, Bovinae Alpha-S2-casein and Beta-casein were also detected, including one peptide sequence for Kappa-casein derived from *Bos*. The individual was recovered from inside a mudbrick substructure, likely the tomb within a pyramid[52].

**New evidence for milk consumption in eastern Africa (Kenya).** Individuals were analysed from six sites in Kenya and milk proteins were identified from three, spanning different time periods and ecological zones. The earliest evidence of milk

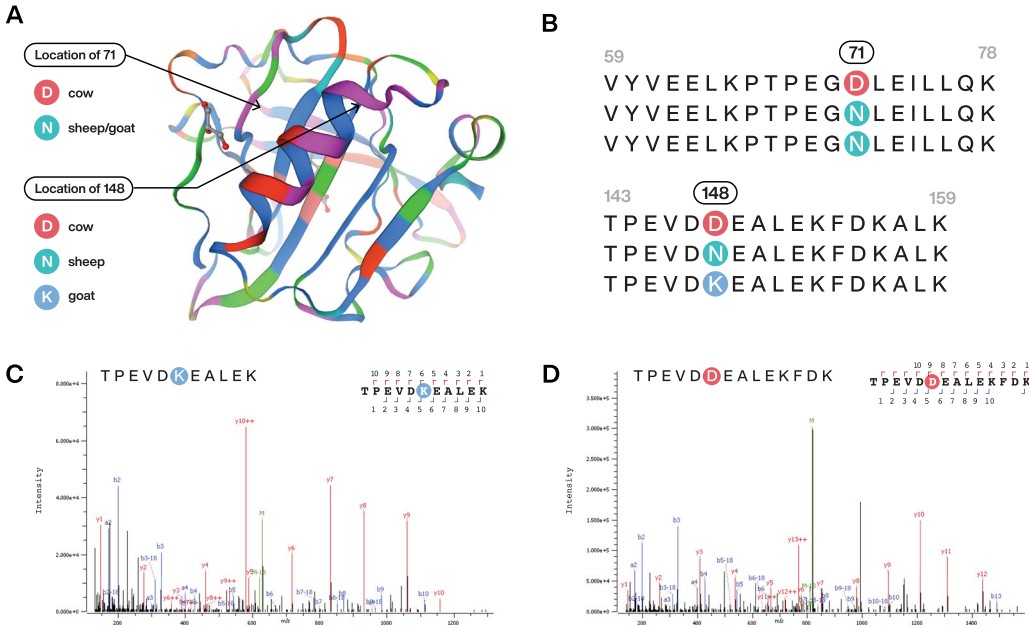

**Fig. 3 Milk β-lactoglobulin proteins identified from ancient dental calculus of individuals from Sudan and Kenya. A** 3D model of β-lactoglobulin showing location of species variant sites. **B** Variations in the amino acid sequence at position 71 and 148 can be used to distinguish between *Bos*, *Ovis* and *Capra*. **C** Spectrum for TPEVDKEALEK specific to *Capra* from individual Z708. **D** Spectrum for TPEVDDEALEKFDK from individual DA323. The deamidation of asparagine results in its conversion to aspartic acid so an unmodified (D) is indistinguishable from (de.N); therefore, this milk peptide is identified as Bovinae/*Ovis*. The 3D image of β-lactoglobulin is from SwissModel [https://swissmodel.expasy.org/repository/uniprot/P02754].

consumption in Kenya was identified at Lukenya Hill, an inselberg in the Athi-Kapiti plains of southern Kenya. Lukenya Hill is associated with several archaeological sites with early evidence of pastoralist material culture and domesticated animal remains[53]. Two samples that yielded results for this study come from locality GvJm202, which is a rockshelter containing human burials attributed to the Pastoral Neolithic phase (~3500–1200 cal. BP)[54] (Supplementary Note 1). Another individual from this locality was directly radiocarbon dated to 3610–3460 cal. BP[31] (Supplementary Note 1 and Supplementary Table 2) confirming early pastoral occupation at GvJm202. Genetic analyses of two individuals from GvJm202 have not identified any known LP-alleles[31].

A second early pastoralist site where BLG proteins were recovered from human dental calculus was Cole's Burial Site (GrJj5a) near Lake Elmenteita within the Central Rift Valley, approximately 150 km northwest of Lukenya Hill[55] (Supplementary Note 1). Dental calculus recovered from one complete human burial yielded BLG peptide sequences belonging to Bovinae/*Ovis* (DA325). This individual is directly radiocarbon dated to 3350–3180 cal. BP (Supplementary Note 1 and Supplementary Table 2)[32], with genetic data demonstrating shared ancestry with other early pastoralist communities, including those at Lukenya Hill[31]. Despite the likely pastoralist diet and lifestyle, and pastoralist ancestry, this individual lacked a genetic signature of derived alleles associated with LP in their genome[32]. Dental calculus was also analysed from an isolated incisor (DA346) from another individual from which BLG peptides (Bovinae/*Ovis*) were identified.

A single individual (DA144) (Skeleton 1) from Molo Cave (GoJi3), also in the Central Rift Valley (Supplementary Note 1), had BLG peptides matching to Bovinae/*Ovis* in their dental calculus. A fragment from the petrous portion of the skull from this individual was directly radiocarbon dated to 1415–1320 cal. BP[31] (Supplementary Note 1 and Supplementary Table 2). The Molo Cave individual dates to very near the transition from the Pastoral Neolithic to the Pastoral Iron Age period in the region and their genetic ancestry[31] reflects relatively late admixture between

indigenous eastern Africa foragers and early herders. Despite the direct evidence for milk consumption, this individual also lacks LP-related alleles[31].

Results of stable isotope analysis for all three sites (Lukenya Hill, Cole's Burial, Molo Cave) (Fig. 4A, Supplementary Note 4, Supplementary Figs. 9–13, Supplementary Data 8 and Supplementary Data 9) support the proteomic findings. Human bone collagen and tooth enamel $\delta^{13}C$ results for the three sites indicate a dietary signal that is aligned with the local environment, mainly $C_4$ grasses, consistent with the consumption of herd animal products (meat or milk). For Lukenya Hill, the average $\delta^{15}N$ for human bone collagen is 12.7‰ compared to 8.0‰ for *Bos* (identified using peptide mass sequencing) (Supplementary Note 4 and Supplementary Data 8), which could indicate the consumption of milk or meat from domesticates. Similarly, the difference between the average $\delta^{15}N$ for humans from Molo Cave (11.6‰) and that of the associated *Bos* and *Capra* specimens (7.0‰) is within the expected range for trophic enrichment (Supplementary Note 4 and Supplementary Data 8), indicating humans were heavily reliant on animal food sources for their protein intake. Overall, $\delta^{13}C$ results for the three Kenyan sites are consistent with people reliant on associated herd animals for meat or milk products[55,56].

Oxygen ($\delta^{18}O$) values for humans ($n = 12$) and faunal ($n = 9$) tooth enamel for all three sites are variable, ranging from −3.7‰ to 0.9‰ and −7.8‰ to 2.2‰, respectively (Fig. 4B and Supplementary Data 9). Of the sites sampled, Molo Cave has the largest faunal isotopic dataset ($n = 6$) including wild taxa (i.e. *Dendrohyrax*, *Heterohyrax*) and domesticates (*Capra* and *Bos*). Caprine $\delta^{18}O$ (−1.5‰ and 1.7‰) are higher than that for the single *Bos* (−4.0‰) sampled from the site. A similar distinction between caprines and cattle has been observed for Pastoral Neolithic sites in Kenya[57] and likely reflects differences in drinking behaviours, the proportions of ingested surface water and plant water, and/or taxonomic differences in fractionation. Given the known multifarious dietary and environmental influences on $\delta^{18}O$ and small number of enamel samples in this

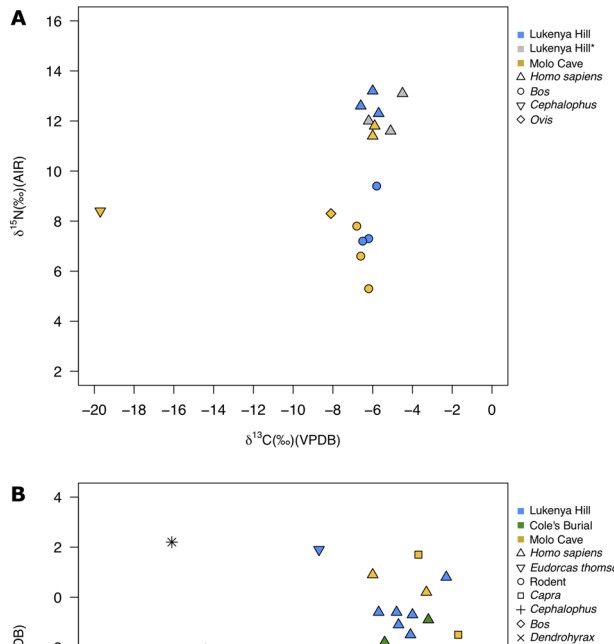

**Fig. 4 Stable isotope results for humans and fauna. A** Bone collagen stable carbon and nitrogen isotope values for Lukenya Hill and Molo Cave. *Values previously published for Lukenya Hill[53]. **B** Tooth enamel stable carbon and oxygen isotope values for Lukenya Hill, Cole's Burial and Molo Cave. The *Cephalophus*, *Dendrohyrax*, and *Heterohyrax* were sampled from Lukenya Hill.

study, we cannot further assess these values, nor those obtained for Lukenya Hill and Cole's Burial, with great confidence. We present them for completeness and future use by other scholars undertaking more detailed $\delta^{18}$O analysis of the assemblages or compiling regional datasets.

## Discussion

The identification of milk proteins in the calculus of eight individuals from five sites across northeastern and eastern Africa provides direct, palaeoproteomic evidence of milk consumption in Africa. Despite the importance of the dietary transition to dairying, and the complex emergence and spread alluded to by current LP allele demography in Africa, understanding of dairying origins in the continent has been impeded by poor preservation of early pastoralist settlements and ancient organic remains. Our provision of direct evidence for the consumption of milk products provides critical insight into the human cultural practices that may have driven selection for diverse LP alleles in Africa.

In this study, milk proteins were detected in calculus from individuals at archaeological sites in Sudan dating to the Neolithic, Kerma, and Meroitic periods. Our results from Kadruka 21 and 1, dating to ~6000 cal. BP and ~4000 cal. BP, respectively, provide direct evidence that milk products were being consumed in the Northern Dongola Reach by at least the sixth millennium BP. This supports previous lipid residue and zooarchaeological evidence from the Neolithic site of Kadero (c. 400 km south of Kadruka) that suggest human use of animal milk by ~6600 cal. BP[38,58]. While much attention has been given to the economic, social, and symbolic significance of cattle in early North Africa due to their predominance in faunal records across the region from the Neolithic onwards[58], our results directly demonstrate, that goats and possibly sheep were also sources of milk products in these early dairying cultures. Obtaining milk from arid-adapted species may have been critical during times of drought in the past as it is among northern African pastoralists today[59,60]. Later in time, the identification of Bovidae and *Bos* milk proteins in calculus from one individual from the Meroitic Period is consistent with textual evidence and rock art images emphasizing the importance of cow's milk at this time in Sudan[61].

The identification of milk proteins in the calculus of individuals from Lukenya Hill and Cole's Burial (~3600–3200 cal. BP) constitutes some of the earliest direct evidence for the arrival of herders in southern Kenya and demonstrates that these groups were already consuming animal milk during the very earliest expansions of herding into the region. Combined with proteomic data from Sudan, and milk lipid residues from one sherd at Dongodien (GaJi4) in northern Kenya by 5000 cal. BP and three sherds at Ngamuriak (GuJf6) and Luxmanda in SW Kenya/ northern Tanzania after c. 3000 cal. BP[40], these data point to milk consumption having been a widespread and persistent component of early herder lifeways. Together, these studies provide strong, multi-proxy evidence that people were regularly relying on access to animal milk throughout the expansion of pastoralism from the Sahara through eastern Africa with no detectable regional or temporal gaps. It is possible that, as has been argued for Mongolia[17,46], caloric and nutritional contributions from animal milk were in fact necessary for the survival of early African herders during expansion across arid regions before widespread plant agriculture (see refs. [62,63]). Increased aridification and drought frequency after ~4500 cal. BP across eastern Africa[64] may have provided the bottleneck effects necessary for rapid selection of LP genes among small populations of herders such that they become evident in ancient genomes by 2100 cal. BP[32]. In such a scenario, an enhanced ability to digest milk products through and after adolescence would have significantly increased an individual's chances of reaching reproductive age.

Our study provides a rare opportunity to link the LP status of specific ancient individuals with direct evidence for their consumption of milk, drawing on combined palaeoproteomic and archaeogenetic investigations. This research clearly demonstrates that ancient African individuals who do not appear to have had a genetic adaptation enabling lactose digestion were nonetheless drinking milk. As noted, not all individuals or populations who drink milk have LP, and ancient dairy consumption may have been enabled by fermentation practices[10,11] or gut microflora[12], as is suggested for populations today. Our study points to the early consumption of dairy having had a role to play in driving selection for LP in eastern Africa. Whether this related to the increased nutritional benefits provided by milk in a particular dietary context, its benefits as a fluid source in arid environments; and/or its utility as a mechanism for withstanding drought or food shortages, for example, remains to be investigated. But it may be that exceptional periods of stress, such as drought, were necessary for driving strong selection for LP. In Africa, drinking milk may have enabled expansion of pastoralists into new regions, and persistence of populations through periods of climatic aridity.

While the proteomic data reported here cannot yet provide a clear explanation for modern LP patterns in Africa, which

undoubtedly reflect palimpsests of migrations across the continent and a potential range of selection pressures, they nonetheless add to a growing body of literature that shows milk consumption began millennia before LP became widespread in several regions of the world. These results show an emerging picture, on a global-scale, of dairying and milk consumption as cultural adaptations that preceded widespread LP, in numerous cases by several millennia. Our results further demonstrate that proteomic evidence for milk consumption has the potential to refine existing narratives for the spread of dairying even in hot environments. Dairy proteins survive in dental calculus from at least 6000 years ago in spite of challenging preservation conditions. Nonetheless, the limited application to date of palaeoproteomic analysis to dental calculus from tropical and arid regions of the world means that extraction methods and data analysis pipelines still need to be optimized for low abundance samples. In order to validate our findings, we created a custom-made OSSD (see Supplementary Note 3 for full details) to investigate if calculus samples matched expected oral protein signatures. However, larger global datasets for ancient dental calculus are needed to fully assess temporal and geographic trends relating to overall preservation[65,66].

Lipid residue analysis has been applied to investigate the inception and development of pastoral economies in northern and eastern Africa, finding that dairy products were used or stored in ceramic vessels[37,38,40,67]. Models for interpreting lipid residue results in European studies[68] may suggest that the very low detection rate for milk lipids from African ceramics indicates minimal contributions of milk to African herder diets. However, proteomic evidence for milk consumption in several individuals supports the idea that the African lipid recovery rates instead reflect preservation biases in arid and tropical climates or that milk was processed in organic containers[69,70]. This reinforces the need for multi-proxy studies that, together, strengthen arguments for widespread use of animal milk among ancient eastern African herders.

Protein-based research opens up the possibility of exploring variation in the relative reliance on the milk of different animal species, and patterns of livestock management, across different regions and populations. The milking of species better suited to arid environments, such as goats, provides potential insight into the adaptations pastoralists developed to survive droughts as well as longer-term climatic aridification. Goats and sheep reproduce much faster following drought events[63] and their role in rapid recovery from climatic stress in the past merits further exploration. Palaeoproteomic analysis also enables dairying to be identified directly, and in specific individuals, with the possibility, as we have demonstrated here, of linking milk consumption to specific genetic ancestries and LP status. In Africa and other parts of the world, preference for storing milk in organic vessels like gourds rather than ceramics[69–71] challenges the use of lipid residue analysis of ceramics as an approach to the study of early milking. Nonetheless, lipid studies remain useful, particularly when skeletal material is unavailable, and in investigations of how fresh milk may have been converted into low lactose products like cheese and yoghurt.

Our data demonstrate that the identification of milk proteins in dental calculus has significant potential to expand our understanding of the global origins and emergence of dairy-based economies, including in warmer and tropical regions of the world. Its ability to identify milk drinking in specific individuals, who can also be assessed using archaeogenetic methods to understand their specific LP status and mutations, offers the opportunity to develop fine-scale proteomic–genomic studies in Africa and elsewhere in future. This study also highlights the strength of combining proteomic and isotopic evidence to provide direct, species-specific information about milk consumption as well as assess human dietary reliance on animal-derived products. Moving forward, the integration of proteomic evidence from dental calculus with the analysis of lipid and protein residues in ceramics, and application of peptide-mass fingerprinting to aid the identification of domesticates, will undoubtedly refine and even reshape our understanding of the emergence and intensification of herding and dairy consumption in Africa and beyond.

## Methods

**Experimental design.** Individuals were analysed from 13 sites across Sudan and Kenya from a range of periods and locations (Supplementary Data 1). All material from Kenya was sampled and exported under permits issued by the National Museums of Kenya. Material from Kadruka 1 and Kadruka 21 was sampled and exported from the Laboratory of Prehistoric Archaeology and Anthropology, University of Geneva under the terms of an agreement with the Section française de la direction des antiquités au Soudan (SFDAS). All other archaeological remains from sites in Sudan were sampled and exported in accordance with section (31A) of the Sudan Antiquities Ordinance 1999. Permission "Ref. NCAM/4/B" was issued by the National Corporation for Antiquities and Museums (NCAM), Khartoum, Sudan.

Fifty-one human dental calculus samples, representing 41 individuals, were analysed for proteins (Supplementary Data 1 and 2), 13 human bones and 21 teeth were analysed using stable isotope analyses ($\delta^{15}$N, $\delta^{13}$C, $\delta^{18}$O) (Supplementary Figs. 9–13, Supplementary Data 8 and Supplementary Data 9), bones from 8 animals were analysed using peptide-mass-fingerprinting to confirm taxonomic identification (Supplementary Table 6 and Supplementary Data 7) and a radiocarbon date was obtained for one human individual (Supplementary Fig. 14).

**Dental calculus sampling.** Dental calculus samples were collected from a number of archaeological remains curated by the National Museums of Kenya, Sudan National Museum and Laboratory of Prehistoric Archaeology and Anthropology Geneva (Supplementary Table 1). Dental calculus was removed from the teeth using a sterile dental scaler and transferred to 2 ml microcentrifuge tubes. Protein extractions were conducted in a clean room facility at the Max Planck Institute for the Science of Human History, Jena.

**Proteomic extraction methods.** Dental calculus samples were extracted using two methods (for discussion of method selection see Supplementary Note 2). Ten samples were extracted using a modified FASP (filter-aided sample preparation) protocol[17,72,73]. Ten to 20 mg of each sample was crushed and briefly predigested in 0.5 M EDTA for 5 min. After the supernatant was removed, the samples were demineralized in 1.0 mL of 0.5 M EDTA for 3–5 days. After demineralization the samples were centrifuged to create a pellet. In all, 10 kDa Millipore Microcon filter units were prepared by adding 50 μL of 8 M urea in 100 mM Tris-HCl (UA). Then 200 μL of EDTA supernatant was transferred to the filter unit and mixed with the UA. The filter unit was centrifuged at $14,000 \times g$ for 10 min. The remaining supernatant was removed from the pellet and stored. Then 30 μL of sodium dodecyl sulfate (SDS)-lysis buffer (4% w/v SDS, 100 mM Tris/HCl pH 8.2, 0.1 M DTT) was added to the pellet and incubated at 95 °C for 5 min followed by centrifugation. After lysis, 200 μL of UA and the supernatant from the lysis were added to the filter unit and mixed. The filter units were centrifuged at $14,000 \times g$ for 20 min and the flow through discarded. The samples were washed with 200 μL of UA followed by centrifugation at $14,000 \times g$ for 20 min.

To alkylate the samples, 100 μL of iodoacetamide (IAA) solution (0.5 M IAA in 8 M UA) was added to the filter units and mixed at 600 r.p.m. for 1 min in the dark followed by incubation in the dark without shaking for 5 min, followed by 15 min centrifugation at $14,000 \times g$. The samples were washed twice with 200 μL of UA, then washed twice with 200 μL of 0.5 M NaCl with centrifugation at $14,000 \times g$ for 20 min at each wash step. One hundred and twenty microlitres of 50 mM triethylammoniumbicarbonate was added to the filters. The samples were digested with 0.4 μg of trypsin overnight at 37 °C and then centrifuged at $14,000 \times g$ for 20 min into a new tube. The flow though was acidified with 5% TFA to a final pH < 2. Stage tips (Thermo Scientific StageTips 200 μL C18 tips) were cleaned with 150 μL 100% methanol, followed by 150 μL 60% acetonitrile (ACN) solution (60% ACN, 0.1% TFA, 39.9% ddH₂O). Stage tips were then equilibrated with two washes of 150 μL of 3% ACN solution (3% ACN, 0.1% TFA, 96.9% ddH₂O). Acidified samples were loaded onto the tips. Tips were then washed twice with 150 μL of 3% ACN solution and the flow through discarded. Peptides were eluted from the stage tip with 150 μL 60% ACN solution into a fresh collection tube, dried in a centrifugal evaporator, and stored at −80 °C ready for LC-MS/MS.

Forty-one samples were extracted using modified Single-pot, solid-phase-enhance sample preparation (SP3) protocol[74]. A 2–10 mg sample of calculus, 3 mg of powered archaeological sheep bone (positive control) and an extraction blank were demineralized in 500 μL of 0.5 M EDTA on a rotator for 3–5 days. After demineralization, samples were centrifuged at $20,000 \times g$ for 10 min and 400 μL of

EDTA supernatant was transferred to a new tube and stored in a −20 ºC freezer as a potential back-up sample. Next, 200 µL 2 M GuHCl was added to each sample and mixed through resuspension. To each sample, 30 µL of 100 mM CAA/100 mM TCEP solution was added to a final concentration of 10 mM CAA/TCEP. Samples were vortexed briefly and then placed in a ThermoMixer for 10 min at 99 °C. After 10 min, samples were removed from the ThermoMixer and left to cool on the bench for 5 min.

Next, 20 µL of prepared Sera-Mag Speedbeads (GE Healthcare) solution (20 µg/µL of 1:1 mixture of hydrophilic:hydrophobic mixture) was added to each sample and mixed through pipetting. A volume of 100% ethanol equal to the total volume (350 µL) was added and each sample briefly shaken to mix. Samples were incubated in a ThermoMixer for 5 min at 1000 r.p.m. at 24 °C. After removal from the ThermoMixer samples were placed in a magnetic rack and left for 1–2 min to allow the beads to migrate to the magnetic wall. The supernatant was removed and stored in a −20 freezer. The beads were then washed three times with 200 µL 80% ethanol, incubated on the magnetic rack for 1–2 min and the supernatant was discarded. The beads were resuspended in 75 µL of 100Mm ammonium bicarbonate and digested with 0.4 µg of trypsin overnight at 37 °C in the ThermoMixer at 750 r.p.m. Samples were centrifuged at 20,000 × g for 1 min and placed in a magnetic rack. Once the beads had migrated to the wall the supernatant was transferred to a new tube. Samples were acidified with 5% TFA to pH < 2. Samples were vortexed briefly to mix and centrifuged. Stage tips (Thermo Scientific StageTips 200 µL C18 tips or three 3M Empore C18 disks placed in 200 µL tips) were cleaned with 150 µL 100% methanol, followed by 150 µL 60% ACN solution (60% ACN, 0.1% TFA, 39.9% ddH2O). Stage tips were then equilibrated with two washes of 150 µL of 3% ACN solution (3% ACN, 0.1% TFA, 96.9% ddH2O). Acidified samples were loaded onto the tips. Tips were then washed twice with 150 µL of 3% ACN solution and the flow through discarded. Tips were then sent for LC-MS/MS analysis to the Functional Genomics Centre Zurich where they were eluted with 150 µL 60% ACN solution. The SP3 protocol can be found on protocols.io [https://doi.org/10.17504/protocols.io.bfgrjjv6].

**LC-MS/MS analysis**. LC-MS/MS was conducted at the Functional Genomics Center Zurich using either a Q-Exactive or a Q-Exactive HF mass spectrometer (Thermo Scientific, Bremen, Germany) equipped with a Digital PicoView source (New Objective) and coupled to a nanoACQUITY or an ACQUITY UPLC M-Class system (Waters AG, Baden-Dättwil, Switzerland), respectively. Solvent composition at the two channels was 0.1% formic acid for channel A and 0.1% formic acid, 99.9% ACN for channel B. Column temperature was 50 °C. For each sample 4 µL of peptides were loaded on a commercial MZ Symmetry C18 Trap Column (100 Å, 5 µm, 180 µm × 20 mm, Waters) followed by nanoEase MZ C18 HSS T3 Column (100 Å, 1.8 µm, 75 µm × 250 mm, Waters). The peptides were eluted at a flow rate of 300 nL/min by a gradient from 8 to 22% B in 49 min, 32% B in 11 min and 95% B in 1 min (Q-Exactive) or from 5 to 40% B in 120 min and 98% B in 5 min (Q-Exactive HF). The column was cleaned after each run with 98% solvent B for 5 min and holding 98% B for 8 min prior to re-establishing loading condition.

The mass spectrometers were operated in data-dependent mode performing HCD (higher-energy collision dissociation) fragmentation on the 12 most intense signals per cycle. The settings were slightly adapted for each instrument. For Q-Exactive analyses, full-scan MS spectra (300–1700 m/z) were acquired at a resolution of 70,000 at 200 m/z after accumulation to a target value (AGC) of 3E6, while HCD spectra were acquired at a resolution of 35,000 using a normalized collision energy of 25 (maximum injection time: 110 ms; AGC 50,000 ions). For Q-Exactive HF analyses, full-scan MS spectra (300–1500 m/z) were acquired at a resolution of 120,000 at 200 m/z after accumulation to a target value (AGC) of 3,000,000, while HCD spectra were acquired at a resolution of 30,000 using a normalized collision energy of 28 (maximum injection time: 50 ms; AGC 10,000 ions). Unassigned singly charged ions and ions were excluded. Precursor masses previously selected for MS/MS measurement were excluded from further selection for 30 s, and the exclusion window was set at 10 ppm. The samples were acquired using internal lock mass calibration on m/z 371.1012 and 445.1200. The mass spectrometry proteomics data were handled using the local laboratory information management system[75].

**Proteomic data analysis**. Authentication of dietary proteins is a major challenge for palaeoproteomic studies of dental calculus. In other studies, covering different regions and time periods, estimation of the deamidation rates of glutamate and asparagine[76,77] has been used to evaluate proteins identified from ancient dental calculus[16,46,78]. These studies compare the deamidation rates of the dietary peptides against those of the oral signature proteins (which are assumed to be authentic[47]). We explored the potential of using deamiDATE[78] to assess deamidation rates of detected milk peptides. However, in our samples the number of deamidation sites in the dairy peptides are too small for statistical comparison (Supplementary Table 3). Noting that the low numbers of detected peptides also hampered using existing strategies focusing on post-translational modifications, we created the Oral Signature Screening Database (OSSD) to assess whether our samples contain ancient proteins representing the oral microbiome. The OSSD cannot verify if food-related proteins are ancient but relies on the assumption that if an oral signature is present, food-related proteins identified from ancient dental calculus are more likely to be endogenous rather than contamination. The OSSD

contains a subset of oral microbes, human inflammatory response proteins, and common contaminants previously reported in ancient dental calculus samples (for full details see Supplementary Note 3). All calculus samples were first processed with the OSSD using Byonic v.3.2.0 (Protein Metrics Inc.)[79] to identify samples that had an oral protein signature in line with previous reports of ancient dental calculus composition (Supplementary Note 3).

The OSSD does not assess the authenticity of a dietary signature. However, it does provide a standardized way of assessing if a calculus sample possesses an oral signature. In this respect, the OSSD is most effective when used as a screening tool for studies focusing on the identification of dietary proteins, rather than those reconstructing the full oral microbiome[47]. For poorly preserved samples, for which other methods of authentication (i.e. deamiDATE[78]) may not be suitable, the OSSD ensures problematic samples are easily identifiable. If dietary proteins are identified in poorly preserved calculus that lacks an oral signature, careful consideration is needed as to whether food-derived proteins are endogenous or modern contamination. In this study, all samples with milk proteins passed the OSSD.

In order to assess other types of damage, for the samples that passed OSSD, the degree to which tryptic vs. non-tryptic cleavage occurred was assessed as a fraction of the total peptides as an additional indication of quality (Supplementary Data 3). While we acknowledge that signatures of age-related damage in ancient dental calculus proteomes are not fully understood, we observe that non-specific cleavage occurs in all of the samples that passed the OSSD with rates of non-tryptic or semi-tryptic cleavage ranging 2.5–24.7% of the peptides in the OSSD searches and 3.4–39.8% of the peptides in the ByonicPreview error tolerant searches (Supplementary Data 3 and Supplementary Data 5). Error-tolerant searches were also performed in Mascot and ByonicPreview against the SwissProt database in order to assess the presence of PTMs across all identified proteins (Supplementary Data 4 and Supplementary Data 5).

Samples with oral signatures were run against SwissProt Release 2019_08 database (560,782 entries) using (1) Byonic (Protein Metrics Inc., Cupertino, California, United States, version 3.2.0) and (2) Mascot (Matrix Science, London, UK, version 2.6.0) and Scaffold (Proteome Software Inc., Portland, Oregon, United States, version 4.9.0) (Supplementary Data 3) using a 95% protein probability cut-off.

**Byonic**. Parameters for Byonic were set as follows: tryptic-specific digestion, a precursor mass tolerance of 5 ppm, a fragment mass tolerance of 0.05 Da, two missed cleavages allowed, carbamidomethyl of cysteine as a fixed modification, variable modifications (2 common, 1 rare) as deamidation of asparagine and glutamate (2 common), oxidation of lysine and methionine (2 common), phosphorylation of serine and threonine (1 common), glutamate or glutamic acid to pryo-glutamate (1 rare), and acetyl at the N terminus (1 rare). Byonic automatically filters the results to show only proteins with less than 1% FDR or up to the 20th decoy, whichever allows more proteins. Protein identifications were filtered to remove any proteins with a log protein p value less than 1 and peptides were filtered to remove any with a score of less than 200. Samples with a high value of non-tryptic cleavage indicated by the OSSD were run using the same parameters with non-specific digestion.

**Mascot and scaffold**. Tandem mass spectra were extracted by MSConvert version 3.0.11781 using the 100 most intense peaks in each spectrum. Charge state deconvolution and deisotoping were not performed. Parameters for Mascot were set as follows: digestion enzyme trypsin, parent ion tolerance of 10 ppm, fragment ion mass tolerance of 0.01 Da, carbamidomethyl of cysteine as a fixed modification, variable modifications of deamidation of asparagine and glutamate, oxidation of lysine and methionine, phosphorylation of serine and threonine, and acetyl at the N terminus.

Scaffold was used to validate MS/MS-based peptide and protein identifications. Peptide identifications were filtered to achieve an FDR less than 1.0% by the Peptide Prophet algorithm[80] with Scaffold delta-mass correction. Protein identifications were accepted if they could be established at an FDR of less than 5.0% and contained at least two unique identified peptides. Final protein and peptide FDR for each sample is listed (Supplementary Data 2). Protein probabilities were assigned by the Protein Prophet algorithm[80]. Proteins that contained similar peptides and could not be differentiated based on MS/MS analysis alone were grouped to satisfy the principles of parsimony. Proteins sharing significant peptide evidence were grouped into clusters. All raw data files and processed files are available through MassIVE repository with accession code MSV000085058.

**Milk peptide identifications**. Milk peptides were individually assessed. A protein–protein alignment search of all peptide spectral matches (PSMs) identified in any sample as from milk proteins (lactoglobulins and caseins) was performed against the entire translated nucleotide database at NCBI using BLAST. Positive peptide identification required 100% homology and 100% coverage of the peptide to the desired dairy protein (Supplementary Data 6). For individuals that passed the OSSD, milk peptides were filtered to achieve a peptide FDR of 0.5% or less in Byonic or a peptide FDR of 1% or less in Mascot/Scaffold (Supplementary Data 3). The differences in FDR scores are due to how each programme assigns and reports

FDR. Individuals were considered to have dairy proteins in their dental calculus if at least four milk PSMs (two unique sequences or all four PSMs starting at position 143) at least two of which were identified in both Mascot and Byonic (see Supplementary Information for more details, Supplementary Figs. 1–7). Non-tryptic peptides were only assessed via the Byonic searches (Supplementary Data 3).

**Stable isotope analysis of bone collagen.** Bone collagen was extracted using a modified Longin[81] method. Approximately 1 g of bone was cleaned using abrasion and demineralized in 0.5 M HCl for 14 days. After demineralization, each sample was rinsed three times with ultra-pure $H_2O$. Samples were gelatinized at 70 °C in pH3 HCl for 48 h and the collagen solution Ezee-filtered. Following 48 h in a freeze dryer, approximately 1.0 mg of collagen was weighed in duplicate into tin capsules for analysis. The $\delta^{13}C$ and $\delta^{15}N$ ratios of the bone collagen were determined using a Thermo Scientific Flash 2000 Elemental Analyser coupled to a Thermo Delta V Advantage mass spectrometer at the Isotope Laboratory, MPI-SHH, Jena. Isotopic values are reported as the ratio of the heavier isotope to the lighter isotope ($^{13}C/^{12}C$ or $^{15}N/^{14}N$) as $\delta$ values in parts per mille (‰) relative to international standards, VPDB for $\delta^{13}C$ and atmospheric $N_2$ (AIR) for $\delta^{15}N$ using the following the equation $[\delta = (R_{sample} - R_{standard})/R_{standard}]$[82]. Results were calibrated against international standards of (IAEA-CH-6: $\delta^{13}C = -10.80 \pm 0.47$‰, IAEA-N-2: $\delta^{15}N = 20.3 \pm 0.2$‰, and USGS40: $\delta^{13}C = -26.38 \pm 0.042$‰, $\delta^{15}N = 4.5 \pm 0.1$‰) and a laboratory standard (fish gelatin: $\delta^{13}C = \sim -15.1$ ‰, $\delta^{15}N = \sim 14.3$‰). Based on replicate analyses long-term machine error over a year is ±0.2‰ for $\delta^{13}C$ and ±0.2‰ for $\delta^{15}N$.

**Stable isotope analysis of tooth enamel.** Teeth were cleaned using air-abrasion and approximately 7 mg enamel was removed from each tooth using diamond-tipped drill bit. For each individual, a bulk sample was taken by sampling enamel from along the full length of the buccal surface. Enamel samples were pretreated with 1 mL 1% NaClO for 1 h and then rinsed with ultra-pure $H_2O$ for a total of three washes, centrifuging each time. Next, 1 mL 0.1 M acetic acid was added for 10 min followed by three more washes with ultra-pure $H_2O$. After the final rinse, each sample was placed in a freeze dryer for 4 h. Alongside the samples of this study, an in-house standard of equid tooth enamel was processed. Approximately 2 mg of each sample was weighed out into a 12 ml borosilicate glass vial. Following reaction with 100% phosphoric acid, gases evolved from the samples were analysed to stable carbon and oxygen isotopic composition using a Thermo Gas Bench 2 connected to a Thermo Delta V Advantage Mass Spectrometer. Carbon ($\delta^{13}C$) and oxygen ($\delta^{18}O$) stable isotope values were calibrated against international standards IAEA NBS 18 ($\delta^{13}C -5.014 \pm 0.032$‰, $\delta^{18}O -23.2 \pm 0.1$‰), IAEA 603 ($\delta^{13}C + 2.46 \pm 0.01$‰, $\delta^{18}O -2.37 \pm 0.04$‰), IAEA CO8 ($\delta^{13}C -5.764 \pm 0.032$‰, $\delta^{18}O -22.7 \pm 0.2$‰), and USGS44 ($\delta^{13}C = \sim -42.1$‰) Precision was assessed by repeat measurements of a laboratory standard (MERCK $CaCO_3$) ($n = 20$, ±0.2‰ for $\delta^{13}C$ and ±0.2‰ for $\delta^{18}O$, $\delta^{13}C = \sim -40.6$‰, $\delta^{18}O = \sim -13.3$‰) Measurement error was ±0.3‰ or less for $\delta^{13}C$ and ±0.2‰ or less for $\delta^{18}O$.

**Morphological identification of faunal remains.** Faunal samples from Lukenya Hill and Molo Cave were identified to element, and when possible, taxon (Supplementary Table 6). Identifications of teeth were made using photographs, drawings, and metric data of eastern African fauna from AJ, as well as published criteria on differentiating caprine teeth[83,84]. Less identifiable fragments were identified to the narrowest taxonomic category possible and size class[85]. Given the fragmentary nature of the postcranial remains, specimens from some sites (Lukenya Hill sites and Molo Cave) could not be identified beyond mammal and size class. These faunal specimens underwent zooarchaeology by mass spectrometry (ZooMS) analysis to refine identifications.

**Zooarchaeology by mass spectrometry.** Highly fragmented faunal bones were analysed by ZooMS to confirm taxonomic identifications (Supplementary Data 7). Samples were reanalyzed following established protocols[86] in which bone samples were demineralized in 0.6 M hydrochloric acid (HCl) for 18 h. The HCl was removed and the sample was rinsed three times in pH 8 solution of 50 mM ammonium bicarbonate (AmBic). After rinsing, the sample was incubated at 70 °C for an hour in 100 μL of 50 mM AmBic. Fifty microlitres of the resulting supernatant was treated with trypsin (Pierce™ Trypsin Protease, Thermo Scientific) at 37 °C for 18 h. Following digestion, the samples were subjected to C18 cleanup (Pierce™ C18 Tips, Thermo Scientific), mixed with a matrix solution of α-cyano-4-hydroxycinnamic of 10 mg/mL in 50% ACN/0.1% trifluoroacetic acid (TFA) and allowed to co-crystallize. Analysis was carried out on an Autoflex MALDI-TOF Bruker Ultraflex II (Bruker Daltonics, Bremen). The resulting mass spectra were screened for diagnostic markers using the FlexAnalysis software and compared against a reference library[87–89] and analysed using mMass. Samples were analysed alongside multiple blanks which all returned negative results and were determined to be empty.

**Radiocarbon dating.** A total of five samples (one bone fragment from petrous portion, three teeth and one hair sample) were sent for radiocarbon dating and were split between the Centre for Isotope Research Groningen (CIO, Lab ID: GrM) and Scottish Universities Environmental Research Centre Radiocarbon Laboratory Glasgow (SUERC, Lab ID: GU). The hair sample from Kadruka 1 SK68 was pretreated with 4% HCl for a short period, rinsed with decarbonated water and dried before combustion. Four teeth were pretreated for dating at SUERC using published methods[90]. A cranial fragment from individual Kadruka 21 Skeleton 129 was pretreated at CIO following established protocols[91] Unfortunately none of the teeth or the cranial bone fragment yielded sufficient collagen for dating. A radiocarbon date was successfully obtained from the hair sample from Kadruka 1 SK68. $^{14}C$ ages were calibrated to calendar years with software programme: OxCal, version 4.3 (ref. [92]), using calibration curve: IntCal13 (ref. [93]) (Supplementary Fig. 14).

**Reporting summary.** Further information on research design is available in the Nature Research Reporting Summary linked to this article.

## Data availability
Raw and processed MS/MS files are available to download via MassIVE repository with accession code MSV000085058. The full Oral Signature Screening Database (OSSD) and associated results are available via MassIVE with accession code MSV000086557 and on the open-access repository Zenodo [https://doi.org/10.5281/zenodo.3698271]. The SP3 protocol is published open-access on protocols.io [https://doi.org/10.17504/protocols.io.bfgrjjv6]. All other supporting data are available within the paper and supplementary information files.

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

## Acknowledgements

The authors are grateful to all local collaborators who contributed to this study. This project was made possible only due to the generous assistance of the staff and curators of the Nairobi National Museum and National Museums of Kenya. All research in Kenya was carried out under permits from the National Commission for Science, Technology and Innovation, Kenya. Sampling and analysis of material from Sudan was made possible only with the assistance of Abdel Rahman Ali Mohamed of the National Corporation for Antiquities and Museums. Our thanks to the staff at the Sudan National Museum, Khartoum and Laboratory of Prehistoric Archaeology and Anthropology, University of Geneva for their help with facilitating the sampling of remains. Thanks also to la Section Française de la Direction des Antiquités au Soudan. The authors express their gratitude to Mary Lucas, Sara Marzo, and Bianca Fiedler for their assistance with isotope sample preparation and mass spectrometry. Our thanks to Mike Dee and Sanne Palstra from the Centre for Isotope Research (CIO), Groningen, and Brian Tripney and Philip Naysmith from the Scottish Universities Environmental Research Centre (SUERC) for radiocarbon analysis. Thanks to Michelle O'Reilly for assistance with figures. This research was supported by the Max Planck Society.

## Author contributions

M. Bleasdale., N.B., A.C., and S.T.G. designed the study. M. Bleasdale, A.J., S.B., J.H., J.Z., and P.R. performed research. M. Bleasdale., K.K.R., A.J., S.B., A.S., J.Z., K.W., S.S., C.T., P.N., J.G., and P.R. analysed data. M. Bleasdale, S.W., J.D., M. Besse, J.R., M.S., H.B., R.C.P., E.N., C.O., F.K.M., M.Z., M.P., and S.T.G contributed material resources. M. Bleasdale., K.K.R., S.T.G., and N.B. wrote the paper.

## Funding

## Competing interests

The authors declare no competing interests.
