## [Peer Review File · Nature Communications]

Reviewers' Comments:

Reviewer #1:

Remarks to the Author:

The paper is clearly written, its abstract reflects the content of the article, and the inferences drawn from the analytic results to be sound.

The findings of the paper are of wide general interest, First, it reports on first attempt to use dental calculus analysis, and specifically proteomics, a technique widely applied to European samples, to recover evidence for milk consumption in northeastern Africa. It establishes that reliable results can be recovered from sub-Saharan calculus samples as old as 6000 years. Second, its results establish that milk consumption was geographically widespread from the dates of the earliest entry of domestic livestock into different parts of this region. Third, in cases where aDNA could be recovered from the same individuals, it establishes that individuals lacking any of the several African mutations for lactase persistence were nonetheless consuming milk. It therefore adds an important geographic dataset to the emerging picture of complexity underpinning dairying and dairy product consumption in various ancient and modern culture.

I will not comment on the methods section of the paper that addresses sampling and proteomic analysis, given that this is not an area of my specialization.

I have only three minor suggestions.

(1) line 272, the reference to Nathan 1996 and Degen 2006 must be put into superscript reference format, as with the balance of the reference.

(2) In discussing aridification and consumption of non-bovine milks (i.e. lines 328-339), the authors might be more specific about which species are more drought-tolerant while still producing milk.

(3) The authors might insert a note for readers not intimately familiar with the record of domestic animals in Africa that, although a mainstay of pastoralist dairying in arid parts of Africa today, the camel was not part of the suite of domestic animals being considered as contributing to the early milk consumption investigated in this paper.

Reviewer #2:

Remarks to the Author:

The manuscript from Bleasdale and colleagues reports the identification of milk peptides from human dental calculus retrieved in eastern Africa. In particular, the authors claim they retrieved dairy proteins in human dental calculus from northeastern Africa, directly demonstrating milk consumption at least six millennia ago. The results are novel and of high interest to others in the community and the wider field, because they contribute to reconstruct the origins of dairying and its relationship to the genetically determined ability to drink milk into adulthood through lactase persistence (LP) in neolithic Africa.

The sample set the authors processed is extremely challenging because the climatic conditions in the areas of origin of the samples are particularly unfavourable to ancient biomolecule preservation. Although the work looks globally convincing, some methodological aspects deserve further consideration.

1- The authors use two different sample preparation methods: FASP and SP3. It would be useful for the authors to compare the performance of the two methods. FASP is the first method used, now a few years ago, for palaeoproteomic analysis of dental calculus (Warinner et al., 2014; PMID: 24562188). Such a method however requires multiple preparation steps that can lead to significant losses. SP3, on the contrary, requires a lower number of preparation steps and does not require transferring the protein extract into multiple tubes. This approach though is not extensively tested on ancient samples and its effectiveness in palaeoproteomics is still unclear. SP3 is based on protein

aggregation and precipitation (Batth et al., 2019, PMID: 30833379), which could be limited or absent when proteins are heavily fragmented. It would be very interesting if the authors could further discuss how effectively SP3 worked on their challenging samples.

2- Due to the heavily degraded nature of the samples, the authors used validation criteria that are below the standards conventionally applied for proteomic analysis of modern samples. Specifically, the authors in some cases identified milk proteins based on the observation of a few MS/MS spectra assigned to a single peptide. Considering the advanced level of protein degradation most probably affecting the samples investigated, such a choice can be considered acceptable. However, to better support the conclusion of the study, the annotated MS/MS spectra defining the key experimental evidence at the foundations of the study should be provided as supplementary information.

3- It looks like the authors matched the MS/MS spectra they experimentally generated only again tryptic peptides. It could be interesting to extend the matching process also to semi-tryptic or non-specific peptides and report the results of these analysis.

4- Although the authors argue that detection of deamidation cannot be used to identify protein damage in this sample set, the dataset generated should be examined, for example with an error-tolerant Mascot search, to detect other forms of protein damage.

Based on the data deposition note, the raw data will be made publicly available and this will allow other researchers to reproduce the work.

Minor points:

Line 61:

Replace: "protein" with "proteins".

Line 92-97:

Improve the readability of the phrase: "Complex patterns such as the presence of high LP frequencies amongst several foraging populations in eastern Africa that do not consume milk, and low LP frequencies amongst several pastoralist populations that regularly do are not understood⁵, while repeated ancient and historical population movements in Africa problematise attempts to assess the selective relationship between LP and environmental context (e.g., drought, UV radiation 23)".
Suggestion: split the phrase into two shorter phrases.

Line 151:

In "from individuals dating to the sixth... ", please specify the exact number of individuals.

Line 237:

In "and their genetic ancestry³⁷. reflects relatively... " remove ".".

Line 454 Proteomic Data Analysis paragraph

For all the software, i.e. Byonic Mascot and Scaffold, used to match spectra to peptide sequences, the match scores used as thresholds to confidently call the matches should be reported.

Line 546:

In: "in 50mM ammonium bicarbonate (AmBic)." Please specify the pH of the solution

Line 548:

In "50ul of..." and similar cases, change to "50µL of..."

Fig. 2B:

Why in the figure for both Kadruka 1 and Kadruka 21 n=4, while in the SI file the authors state: "For

this study a selection of skeletal remains (n=10) were sampled, five each from Kadruka 1 (KDK 1) and Kadruka 21 (KDK 21) representing both the Neolithic and Kerma periods."? Is the number of samples selected from each site 4 or 5? Please cross check and if necessary correct.

Enrico Cappellini

Reviewer #3:

Remarks to the Author:

This paper reports novel palaeoproteomic results from east African prehistoric dental calculus samples. Although a large number of samples were analysed, milk proteins were only found in a small proportion of these (8 out of 41 individuals), but the paper still provides an interesting addition to our growing knowledge of prehistoric dairy consumption and proteomic preservation. The paper is well written, and will be of interest to palaeoproteomic researchers, prehistoric archaeologists, and those with an interest in the origins of dairy consumption. Whilst I recommend the paper for publication, I would suggest that a number of issues need to be addressed prior to publication.

1. Assessment of authenticity and deamidation patterns

The major issue I can see with the paper revolves around the approach taken to determine the authenticity of milk peptides recovered. The determination that proteins recovered from archaeological materials are indeed ancient and endogenous, rather than modern contaminants, is of utmost importance. Most palaeoproteomic studies now commonly utilise deamidation patterns, normally of asparagine and glutamine, to determine that the peptides recovered conform with damage expected from ancient proteins. The authors here have instead adopted a different approach to authenticating their data, through the use of a new Oral Signature Screening Database (OSSD). The discussion of the nature of this approach, the reasons behind why it was adopted, and what the database contains is important and highly significant – and underlies all data presented within the manuscript. With this in mind, I believe it warrants inclusion within the main manuscript of the paper, rather than in the Supplementary Information (currently pages 5-8). Whilst I acknowledge it will not be possible to include all of the text within the main manuscript, a more detailed explanation of the approach is needed than is currently presented in lines 160-166 of the manuscript.

Whilst I acknowledge the authors' comments surrounding the reasons as to why this approach was adopted (Supplementary Information page 5), the major issue I can see with this approach however is that whilst it authenticates the oral signature of the calculus samples, it does not authenticate the individual milk proteins. This therefore means that a sample with an endogenous oral signature may still contain contaminant milk proteins. The authors themselves do acknowledge this on page 8 of the Supplementary Information.

As there are now a number of published methods for assessing levels of deamidation (e.g. Mackie et al. 2018; Ramsøe et al. 2020), it would be highly beneficial if the authors could run this analysis of deamidation patterns on their data. This additional analysis would not be difficult to undertake, but will add significantly to the paper. As the paper claims the first proteomic evidence for milk consumption in prehistoric Africa, it is important to fully determine that these milk peptides are indeed ancient.

2. Introduction

Although I welcome the introduction and thorough discussion of the emergence of pastoralism, LP, and milk consumption within eastern Africa, I find that some of the paragraphs within the introductory

section of the paper could benefit from some re-working.

In particular, the paragraph starting line 84 refers to a "growing corpus of aDNA studies", but does not reference these. How many prehistoric human skeletal remains have been analysed for LP thus far? A more explicit discussion of the aDNA work already undertaken is needed here, particularly as this has implications for the claims made on line 99 that the earliest evidence of LP in Africa is c.6ky after the emergence of pastoralism. Additionally, the sentence starting on line 92 is over-long and not particularly clear, and it would be beneficial to re-write this. The reference to a "selective relationship between LP and environmental context" on line 96 could be also be expanded on.

Given that the milk proteins found within the individuals studied here were BLG and caseins, it would also be beneficial to provide some information on these proteins within the introductory section of the paper.

3. Isotopic analyses

The generation and inclusion of stable isotope data within the paper is notable. However, as the isotopic results are briefly discussed in lines 240-251 of the manuscript, it would be good to include an isotope plot of all $\delta^{13}\text{C}$ and $\delta^{15}\text{N}$ values (human and fauna) in the main text of manuscript.

Line 133 of the text also mentions $\delta^{18}\text{O}$ isotopic analysis was undertaken, however the results of this data are not discussed anywhere in the manuscript text. This analysis is also mentioned in section 3 of the Supplementary Information, but it is not clear exactly which individuals this was undertaken on, and no interpretation of the data generated is provided (as is done with the $\delta^{13}\text{C}$ and $\delta^{15}\text{N}$ data). Inclusion of this information is therefore needed.

4. Milk processing

With regards to potential consumption of different dairy products, the Discussion section of the paper mentions previous organic residue analysis undertaken, and the recovery of dairy lipids – do any of these studies show the presence thermally modified compounds, such as long-chain ketones? If so, this would potentially indicate processed milk products were being produced. It is also not clear from the discussion (paragraph starting line 317) whether the previous organic residue analysis work was undertaken on material from the same sites which were sampled for calculus here – or indeed if the studies were focused on East African material.

Minor comments:

- In the caption of Figure 2 it would be good to include a note that the pie charts represent the number of individuals with the presence of milk proteins.
- Line 192: need to specify here which milk proteins were detected
- Line 231: should read "BLG peptides" not just "BLG"
- Line 272: These references have not been included correctly and are missing from the reference list at the end of the manuscript
- Line 412: "stored -20 freezer" should read as "stored in a -20°C freezer"
- Supplementary Information, page 5, paragraph 3: "aparagene" should read "asparagine"

- Throughout the text (both the manuscript and the supplementary information) there appears to be a mix of UK English and American English spellings (e.g. demineralised and demineralized are both used, as are analysed and analyzed)

We would like to thank the Reviewers for their highly constructive comments. We are glad that they all recognised the importance of our data and conclusions. We thank them also for their detailed suggestions to improve our manuscript. We have responded to these on a point-by-point basis below as well as in the manuscript word document with tracked changes.

REVIEWER COMMENTS

Reviewer #1 (Remarks to the Author):

The paper is clearly written, its abstract reflects the content of the article, and the inferences drawn from the analytic results to be sound.

The findings of the paper are of wide general interest, First, it reports on first attempt to use dental calculus analysis, and specifically proteomics, a technique widely applied to European samples, to recover evidence for milk consumption in northeastern Africa. It establishes that reliable results can be recovered from sub-Saharan calculus samples as old as 6000 years. Second, its results establish that milk consumption was geographically widespread from the dates of the earliest entry of domestic livestock into different parts of this region. Third, in cases where aDNA could be recovered from the same individuals, it establishes that individuals lacking any of the several African mutations for lactase persistence were nonetheless consuming milk. It therefore adds an important geographic dataset to the emerging picture of complexity underpinning dairying and dairy product consumption in various ancient and modern culture.

We thank the Reviewer for their positive comments in relation to the importance of our manuscript and its contribution of much needed direct evidence of milk consumption in ancient Africa.

I will not comment on the methods section of the paper that addresses sampling and proteomic analysis, given that this is not an area of my specialization.

I have only three minor suggestions.

(1)line 272, the reference to Nathan 1996 and Degen 2006 must be put into superscript reference format, as with the balance of the reference.

Corrected.

(2)In discussing aridification and consumption of non-bovine milks (i.e. lines 328-339), the authors might be more specific about which species are more drought-tolerant while still producing milk.

Lines 388-392 now read: *“The milking of species better suited to arid environments, such as goats, provides potential insight into the adaptations pastoralists developed to survive droughts as well as longer-term climatic aridification. Goats and sheep reproduce much faster following drought events⁶³ and their role in rapid recovery from climatic stress in the past merits further exploration.”*

(3) The authors might insert a note for readers not intimately familiar with the record of domestic animals in Africa that, although a mainstay of pastoralist dairying in arid parts of Africa today, the camel was not part of the suite of domestic animals being considered as contributing to the early milk consumption investigated in this paper.

We thank the Reviewer for making this interesting point; however, we do not think this is necessary since camel milk was not identified in our results.

Reviewer #2 (Remarks to the Author):

The manuscript from Bleasdale and colleagues reports the identification of milk peptides from human dental calculus retrieved in eastern Africa. In particular, the authors claim they retrieved dairy proteins in human dental calculus from northeastern Africa, directly demonstrating milk consumption at least six millennia ago. The results are novel and of high interest to others in the community and the wider field, because they contribute to reconstruct the origins of dairying and its relationship to the genetically determined ability to drink milk into adulthood through lactase persistence (LP) in neolithic Africa.

The sample set the authors processed is extremely challenging because the climatic conditions in the areas of origin of the samples are particularly unfavourable to ancient biomolecule preservation. Although the work looks globally convincing, some methodological aspects deserve further consideration.

We thank the Reviewer and are glad that they recognise the novelty of our findings and importance of our data for furthering our understanding of dairying and lactase persistence. With regards to their concerns in relation to the methods we agree that some aspects need further development and have addressed their specific points below.

1- The authors use two different sample preparation methods: FASP and SP3. It would be useful for the authors to compare the performance of the two methods. FASP is the first method used, now a few years ago, for palaeoproteomic analysis of dental calculus (Warinner et al., 2014; PMID: 24562188). Such a method however requires multiple preparation steps that can lead to significant losses. SP3, on the contrary, requires a lower number of preparation steps and does not require transferring the protein extract into multiple tubes. This approach though is not extensively tested on ancient samples and its effectiveness in palaeoproteomics is still unclear. SP3 is based on protein aggregation and precipitation (Batth et al., 2019, PMID: 30833379), which could be limited or absent when proteins are heavily fragmented. It would be very interesting if the authors could further discuss how effectively SP3 worked on their challenging samples.

We agree entirely with the Reviewer about both the difficulty of sample loss when using FASP and the untested potential issues of using SP3 for ancient samples. We have added the citation mentioned (Batth et al., 2019) and included more detailed discussion of FASP and SP3 in a new section in the Supplementary “Proteomic Extraction Methods” which reads as follows:

“Two proteomic extraction methods were used in this study: FASP and SP3. Modified FASP (Filter-Aided Sample Preparation)¹⁷ protocols have successfully been applied in several

ancient dental calculus studies¹⁸⁻²⁰, including the first study to report milk peptides²¹. However, filter-based protocols are less suited to small amounts of starting material²². Recently, Single-pot, solid-phase-enhance sample preparation (SP3) has been developed to overcome sample size limitations of FASP^{23,24}. However, this method is based on protein precipitation and aggregation²⁵, which could be limited or absent in cases where proteins are heavily fragmented. For a full description of SP3, see methods section of manuscript and protocol published on [protocols.io](https://www.protocols.io) DOI: [dx.doi.org/10.17504/protocols.io.bfgrijv6](https://doi.org/10.17504/protocols.io.bfgrijv6).”

Since this is the first use of SP3 on ancient dental calculus, we have also published our full protocol on [protocols.io](https://www.protocols.io) ([dx.doi.org/10.17504/protocols.io.bfgrijv6](https://doi.org/10.17504/protocols.io.bfgrijv6)) which will be publically accessible once this paper is published.

We also agree that comparison between the methods would be useful. However, due to starting sample sizes, it was only possible to prepare samples from 10 individuals with both SP3 and FASP. Due, largely to preservation issues in these samples, only 2 of these individuals had oral signatures from both FASP and SP3. In addition, 2 individuals had oral signatures only from FASP and 2 only from SP3. Due to the small sample size and possible issues with differential and poor preservation, we feel that our data cannot be used on its own to compare protein recovery for the two different methods. Therefore, we do not include a detailed comparison here.

In our case, the choice to continue with SP3, was based on the small quantity of starting material for the majority of the samples. Unfortunately there was only enough material to do one SP3 extraction in many cases.

We have added the following to the Supplementary:

“In this study, dental calculus samples from Kadruka (n = 10) were extracted using both methods. For other sites, sample were extracted only with SP3 because start weights were too low for FASP. While this is the first study using SP3 on archaeological dental calculus and differences were observed between the methods, statistical comparisons were not conducted due to the small number of samples prepared with both methods.”

2- Due to the heavily degraded nature of the samples, the authors used validation criteria that are below the standards conventionally applied for proteomic analysis of modern samples. Specifically, the authors in some cases identified milk proteins based on the observation of a few MS/MS spectra assigned to a single peptide. Considering the advanced level of protein degradation most probably affecting the samples investigated, such a choice can be considered acceptable. However, to better support the conclusion of the study, the annotated MS/MS spectra defining the key experimental evidence at the foundations of the study should be provided as supplementary information.

We are glad that the Reviewer acknowledges the challenges of working with degraded materials retrieved from archaeological contexts in Africa. We are grateful for their suggestion to include annotated spectra for the milk peptides as this aids our justification of the criteria used in our study. The annotated spectra have now been added to the supplementary (Supplementary Figs. 1-8).

3- It looks like the authors matched the MS/MS spectra they experimentally generated only against tryptic peptides. It could be interesting to extend the matching process also to semi-tryptic or non-specific peptides and report the results of these analysis.

The searches with the OSSD database which contained the dairy proteins, were done with non-specific digestion. The proportion of tryptic and non-tryptic peptides found in these searches are reported in Dataset S3. Three of the ten individuals which had milk proteins have peptides with non-specific digestion (DA351, Z708, and DA156) and these are reported in individuals in Dataset S6. One of the samples (DA351) only has non-specific tryptic peptides. However, this individual was run with both FASP and SP3 extractions. In the FASP extraction only tryptic LACB peptides were identified. In the SP3 extraction only non-tryptic CASB peptides were identified. Although this could indicate interesting information about the biases with FASP and SP3, for the reasons outlined above, we do not feel that our data supports making these comparisons.

Since the non-tryptic searches did not result in any additional individuals having milk proteins, but only provided additional support for dairy, we only briefly mentioned this in the manuscript text.

We have now added additional lines (550-559) to the Methods:

“In order to assess other types of damage, for the samples that passed OSSD, the degree to which tryptic vs. non-tryptic cleavage occurred was assessed as a fraction of the total peptides as an additional indication of quality (Supplementary Dataset 3). While we acknowledge that signatures of age-related damage in ancient dental calculus proteomes are not fully understood, we observe that non-specific cleavage occurs in all of the samples that passed the OSSD with rates of non-tryptic or semi-tryptic cleavage ranging 2.5-24.7% of the peptides in the OSSD searches and 3.4-39.8% of the peptides in the ByonicPreview error tolerant searches (Supplementary Dataset 3, Supplementary Dataset 5).”

4- Although the authors argue that detection of deamidation cannot be used to identify protein damage in this sample set, the dataset generated should be examined, for example with an error-tolerant Mascot search, to detect other forms of protein damage.

We have now provided the results of the error-tolerant searches in Mascot and Byonic in Dataset S4-5.

And added the following line to the Methods:

“Error-tolerant searches were also performed in Mascot and ByonicPreview against the SwissProt database in order to assess the presence of PTMs across all identified proteins (Supplementary Datasets 4-5).”

Other modifications are found in our samples, but as we have no modern comparison for dental calculus from this area of the world, we are hesitant to conclude that any of these modifications are indicators of age related damage.

Based on the data deposition note, the raw data will be made publicly available and this will allow other researchers to reproduce the work.

The raw files and results from the OSSD searches for all samples in Byonic as well as the swissprot searches in both Byonic and MASCOT have been uploaded to massIVE. Full details are given in the manuscript and Supplementary. Additionally an accompanying article on the OSSD is published on Zenodo.

Minor points:

Line 61:

Replace: "protein" with "proteins".

Corrected.

Line 92-97:

Improve the readability of the phrase: "Complex patterns such as the presence of high LP frequencies amongst several foraging populations in eastern Africa that do not consume milk, and low LP frequencies amongst several pastoralist populations that regularly do are not understood⁵, while repeated ancient and historical population movements in Africa problematise attempts to assess the selective relationship between LP and environmental context (e.g., drought, UV radiation ²³).". Suggestion: split the phrase into two shorter phrases.

We have followed the Reviewer's very useful advice to divide the sentence into two shorter ones, and agree that this makes this section more readable.

Line 151:

In "from individuals dating to the sixth... ", please specify the exact number of individuals.

Corrected to: *"The earliest milk peptides recovered are from Sudan, from a c. 6th millennium BP individual; milk was also identified as early as the fourth millennium BP in Kenya."*

Line 237:

In "and their genetic ancestry³⁷. reflects relatively... " remove ".".

Corrected.

Line 454 Proteomic Data Analysis paragraph

For all the software, i.e. Byonic Mascot and Scaffold, used to match spectra to peptide sequences, the match scores used as thresholds to confidently call the matches should be reported.

"Samples with oral signatures were run against SwissProt Release 2019_08 database (560,782 entries) using 1) Byonic (Protein Metrics Inc., Cupertino, California, United States, version 3.2.0) and 2) Mascot (Matrix Science, London, UK, version 2.6.0) and Scaffold (Proteome Software Inc., Portland, Oregon, United States, version 4.9.0) (Supplementary Dataset 3) using a 95% protein probability cut off."

We have also added details to the "Byonic" paragraph of the Methods section:

"Protein identifications were filtered to remove any proteins with a log protein p-value less than 1 and peptides were filtered to remove any with a score of less than 200. Samples with

a high value of non-tryptic cleavage indicated by the OSSD were run using the same parameters with non-specific digestion.”

Line 546:

In: "in 50mM ammonium bicarbonate (AmBic)." Please specify the pH of the solution

The line now reads: *“...in a pH 8 solution of 50mM ammonium bicarbonate (AmBic)”*.

Line 548:

In "50ul of..." and similar cases, change to "50µL of..."

Corrected.

Fig. 2B:

Why in the figure for both Kadruka 1 and Kadruka 21 n=4, while in the SI file the authors state: "For this study a selection of skeletal remains (n=10) were sampled, five each from Kadruka 1 (KDK 1) and Kadruka 21 (KDK 21) representing both the Neolithic and Kerma periods."? Is the number of samples selected from each site 4 or 5? Please cross check and if necessary correct.

Thank you for catching the discrepancy. Figure 2B contains pie charts which show the individuals (now corrected) that have dairy peptides out of the total that passed the OSSD. Individuals from Kadruka had their calculus extracted with both FASP and SP3. These results have now been combined to represent individuals in this chart. This is why in some places the numbers for samples and individuals are not the same. For Kadruka we have been careful to describe the difference between individuals and samples where necessary.

The Figure caption now reads: *“Map of sites with calculus containing milk proteins. (A) Area of study in relation to the spread of cattle-based pastoralism across Africa (after ³⁵). (B) Pie charts showing the number of individuals per site with milk proteins (shaded) proportionate to the total number of individuals that passed screening with Oral Signature Screening Database (see Methods and Supplementary Information for full details). Neolithic: ~8000-5500 cal. BP; Kerma: ~4450-3450 cal. BP; Pastoral Neolithic: ~3500-1200 cal. BP; Meroitic: ~2300–1600 cal. BP.”*

Enrico Cappellini

Reviewer #3 (Remarks to the Author):

This paper reports novel palaeoproteomic results from east African prehistoric dental calculus samples. Although a large number of samples were analysed, milk proteins were only found in a small proportion of these (8 out of 41 individuals), but the paper still provides an interesting addition to our growing knowledge of prehistoric dairy consumption and proteomic preservation. The paper is well written, and will be of interest to palaeoproteomic researchers, prehistoric archaeologists, and those with an interest in the origins of dairy consumption. Whilst I recommend the paper for publication, I would suggest that a number of issues need to be addressed prior to publication.

We thank the Reviewer for acknowledging the cross-disciplinary importance of our research and novelty of our findings despite the relatively small number of positive results.

1. Assessment of authenticity and deamidation patterns

The major issue I can see with the paper revolves around the approach taken to determine the authenticity of milk peptides recovered. The determination that proteins recovered from archaeological materials are indeed ancient and endogenous, rather than modern contaminants, is of utmost importance. Most palaeoproteomic studies now commonly utilise deamidation patterns, normally of asparagine and glutamine, to determine that the peptides recovered conform with damage expected from ancient proteins. The authors here have instead adopted a different approach to authenticating their data, through the use of a new Oral Signature Screening Database (OSSD). The discussion of the nature of this approach, the reasons behind why it was adopted, and what the database contains is important and highly significant – and underlies all data presented within the manuscript. With this in mind, I believe it warrants inclusion within the main manuscript of the paper, rather than in the Supplementary Information (currently pages 5-8). Whilst I acknowledge it will not be possible to include all of the text within the main manuscript, a more detailed explanation of the approach is needed than is currently presented in lines 160-166 of the manuscript.

We fully support the Reviewer's comment that issues of authenticity are one of the major challenges for proteomic research of ancient dental calculus. As the first study to analyse calculus from prehistoric contexts in Africa we were careful to explore and develop new measures to authenticate our findings. Thank you for your comments on the balance of what information is useful in the manuscript as opposed to the SI.

We now have added the following to the Methods:

“Authentication of dietary proteins is a major challenge for palaeoproteomic studies of dental calculus. In other studies, covering different regions and time periods, estimation of the deamidation rates of glutamate and asparagine^{76,77} has been used to evaluate proteins identified from ancient dental calculus^{16,46,78}. These studies compare the deamidation rates of the dietary peptides against those of the oral signature proteins (which are assumed to be authentic⁴⁷). We explored the potential of using deamiDATE⁷⁸ to assess deamidation rates of detected milk peptides. However, in our samples the number of deamidation sites in the dairy peptides are too small for statistical comparison (Supplementary Dataset 7). Noting that the low numbers of detected peptides also hampered using existing strategies focusing on post-translational modifications, we created the Oral Signature Screening Database (OSSD) to assess whether our samples contain ancient proteins representing the oral microbiome. The OSSD cannot verify if food-related proteins are ancient but relies on the assumption that if an oral signature is present, food-related proteins identified from ancient dental calculus are more likely to be endogenous rather than contamination. The OSSD contains a subset of oral microbes, human inflammatory response proteins and common contaminants previously reported in ancient dental calculus samples (for full details see Supplementary Information). All calculus samples were first processed with the OSSD using Byonic v.3.2.0 (Protein Metrics Inc.)⁷⁹ to identify samples that had an oral protein signature in line with previous reports of ancient dental calculus composition (Supplementary Information).”

Whilst I acknowledge the authors' comments surrounding the reasons as to why this approach was adopted (Supplementary Information page 5), the major issue I can see with

this approach however is that whilst it authenticates the oral signature of the calculus samples, it does not authenticate the individual milk proteins. This therefore means that a sample with an endogenous oral signature may still contain contaminant milk proteins. The authors themselves do acknowledge this on page 8 of the Supplementary Information.

As there are now a number of published methods for assessing levels of deamidation (e.g. Mackie et al. 2018; Ramsøe et al. 2020), it would be highly beneficial if the authors could run this analysis of deamidation patterns on their data. This additional analysis would not be difficult to undertake, but will add significantly to the paper. As the paper claims the first proteomic evidence for milk consumption in prehistoric Africa, it is important to fully determine that these milk peptides are indeed ancient.

We thank the Reviewer for raising this important point about authenticating the individual milk proteins. Sadly all current methods for authentication (including deamidation) are based on having statistical power and rarely can authenticate a single peptide, but rather authenticate a collection of peptides. Both with the OSSD method and deamidation any individual peptide could still be derived from contamination. In the case of the OSSD, as the Reviewer correctly states, a sample with endogenous oral signature could still have milk proteins that derived from contamination. We acknowledge this in the SI, but have now included it in the main text to be more clear:

(Lines 541-549) *“The OSSD does not assess the authenticity of a dietary signature. However, it does provide a standardised way of assessing if a calculus sample possesses an oral signature. In this respect, the OSSD is most effective when used as a screening tool for studies focusing on the identification of dietary proteins, rather than those reconstructing the full oral microbiome⁴⁷. For poorly preserved samples, for which other methods of authentication (i.e. deamiDATE⁷⁸) may not be suitable, the OSSD ensures problematic samples are easily identifiable. If dietary proteins are identified in poorly preserved calculus that lacks an oral signature, careful consideration is needed as to whether food-derived proteins are endogenous or modern contamination. In this study, all samples with milk proteins passed the OSSD.”*

We also recognise the value of using deamidation rates as a way of assessing whether proteins are truly ancient. In a sample with a small number of proteins which have high coverage (such as bone collagen), deamidation can provide independent authentication of age when used against a comparative dataset of bones. However, for dental calculus, deamidation has been used only comparatively to assess if the deamidation rates in the oral signature and the dietary signature are similar. This makes the assumption that the oral signature is “ancient” and the dietary signature is tested against it for contamination.

When looking at the dairy peptides in our samples many of them have too few sites of deamidation to reliably use for authentication. We have provided a new table (Supplementary Dataset 7) that indicates for each sample the number of possible sites of deamidation across all dairy peptides and how many are deamidated. In LACB, there are two frequently observed SAPs where the options are N or D. In our samples all observations of these sites were deamidated (D). However, there is no way to determine this represents the original sequence or a deamidation so these are reported separately in the table and not included in the discussion. Out of the eleven samples with dairy peptides (10 individuals) these are the results of the possible deamidation sites in all dairy peptides:

5 samples - no possible deamidation sites

1 sample - 1 possible deamidation site, 0 deamidated

2 samples - 2 possible deamidation sites, 0 deamidated

1 sample - 3 possible deamidation sites, 3 deamidated

1 sample - 4 possible deamidation sites, 3 deamidated

1 sample - 17 possible deamidation sites, 1 deamidated

The samples with two or fewer possible deamidation sites are difficult or impossible to say anything using deamidation rates. The two samples with 3 and 4 possible sites are highly deamidated, but there are still too few possible deamidation sites to reliably do a statistical comparison.

The final sample (DA156) is the only sample where there are enough possible sites for any statistical comparison. This sample only has 1/17 possible sites deamidated. However, this sample has more peptides corresponding to dairy and oral signature many of which are caseins. The individual was excavated from the Berber Meroitic Cemetery and is believed to date to ~2000 cal. BP, making the calculus sample one of the youngest in this study to have dairy peptides. With the exception of Molo Cave, all other samples with milk proteins are estimated to be at least 1,000 years older than the sample from Berber, with samples from Kadruka 21 potentially 4,000 years older. While other individuals were excavated from relatively closed settings (i.e. burial mounds (Kadruka), cave (Molo) rockshelter (Lukenya Hill)) the Berber individual is the only to have been enclosed in a mudbrick structure. Finally, as seen in the error tolerant search results (Supplementary Datasets 4-5) DA156 has less overall deamination than the other samples in both MASCOT and Byonic. Taken into account all these factors we therefore expect better preservation in this individual.

Deamidation rates correlate with age, but also depend on storage, method of sample processing, and the location in the peptide and protein. Most of the detailed work on this has been done on collagen and without suitable numbers of deamidation sites in the dairy peptides that allow for statistical analysis, the current tools are not suitable. Finally, even with these current tools there is still no way to authenticate an individual peptide. We have provided all of the different cutoffs and approaches used to authenticate our data in order for others to critically evaluate our methods.

2. Introduction

Although I welcome the introduction and thorough discussion of the emergence of pastoralism, LP, and milk consumption within eastern Africa, I find that some of the paragraphs within the introductory section of the paper could benefit from some re-working.

We have edited several sentences in our Introduction to improve clarity.

In particular, the paragraph starting line 84 refers to a “growing corpus of aDNA studies”, but does not reference these. How many prehistoric human skeletal remains have been analysed for LP thus far? A more explicit discussion of the aDNA work already undertaken is needed here, particularly as this has implications for the claims made on line 99 that the earliest evidence of LP in Africa is c.6ky after the emergence of pastoralism.

We have now added the appropriate references and explained why it is challenging to report the number of individuals tested for LP so far.

The paragraph now reads:

“The growing corpus of aDNA research for Africa, which has generated genetic data for over 100 individuals²⁴⁻³³ has so far identified only one LP-associated allele in a single individual from northern Tanzania (~2000 cal. BP)³². Establishing the true frequency of LP in ancient Africa is constrained by the relatively small ancient genomic dataset for the continent. In addition, the absence of a LP-associated allele in some individuals can be attributed to poor coverage on LP-associated genomic regions rather than a true absence of the LP-related genetic signature. Nevertheless, current available evidence places LP in Africa ~6000 years after the introduction of livestock to the continent, and ~1000 years after the development of specialized cattle, sheep and goat herding economies in eastern Africa^{34,35}. As in Europe, higher frequencies of LP may have emerged relatively late and well after the beginnings of dairy consumption in Africa, though more direct insight into the chronology of early milk drinking and high-quality ancient LP data is needed to test this.”

While we agree it would be useful to give exact numbers here, we feel this would warrant detailed explanation of the complexities of reporting this data. For example, it depends on how readers define Africa (i.e. mainland Africa with/without neighbouring islands) whether all LP-associated sites on the genome were sequenced, whether the sequenced ancient genome is of good quality etc. Furthermore, not all published studies include whole genome sequence data, which has more LP-associated sites sequenced than capture sequence data (see Vicente and Schlebusch 2020 for review of African aDNA studies). Most importantly, poor data quality because of poor preservation could result in the absence of LP-associated alleles in the genome. Collectively, these issues make it problematic to report the total number of ancient individuals “tested” for LP in comparison to the single individual currently with LP-associated alleles. We feel that discussing all these nuances would distract from the main focus of our study. We have therefore included a brief explanation of how the absence of LP alleles can result from missing data, and meanwhile directed the reader to all the relevant aDNA studies, including a new review of aDNA data from Africa (Vicente and Schlebusch 2020). We hope these changes are satisfactory.

Additionally, the sentence starting on line 92 is over-long and not particularly clear, and it would be beneficial to re-write this.

The section now reads:

“Despite this, the historical relationship between LP selection and milking in Africa is vastly under-studied. Counterintuitive patterns, such as the persistence of high LP frequencies amongst foraging populations in eastern Africa that are not assumed to have deep histories of animal management or milk consumption (e.g. Ogiek, Sandawe, Hadza), and low LP frequencies amongst pastoralist populations that regularly consume dairy products (e.g. Dinka), are not fully understood^{4,5,22}.”

The reference to a “selective relationship between LP and environmental context” on line 96 could be also be expanded on.

We thank the reviewer for their suggestion but we do not think this is necessary in the Introduction.

Given that the milk proteins found within the individuals studied here were BLG and caseins, it would also be beneficial to provide some information on these proteins within the introductory section of the paper.

We have added the following line to the Introduction:

“Typically, these studies have identified the milk whey protein beta-lactoglobulin as opposed to other proteins more abundant in whole milk, such as caseins, although the reason for this detection bias is not yet well-understood⁴⁵.”

BLG is the most frequently detected milk protein in proteomic studies of ancient dental calculus but is not the most abundant protein found in whole milk (see Hendy et al. 2018 .Proc. R. Soc. B285:20180977). Alpha-S1-casein (CASA1) is the most abundant protein in whole milk yet it was not detected in this study and is generally found at far lower frequencies than BLG in ancient calculus studies. The reason for this discrepancy is still unknown but could be due to the chemical properties of BLG meaning it preserves better, or due to biases in extraction method or data analysis. More work is required to identify potential biases in recovery of milk proteins. We feel it would not be appropriate to discuss further the types of milk protein (i.e. BLG vs caseins) found since we cannot say with certainty how this reflects the consumption of different quantities of milk products etc.

3. Isotopic analyses

The generation and inclusion of stable isotope data within the paper is notable. However, as the isotopic results are briefly discussed in lines 240-251 of the manuscript, it would be good to include an isotope plot of all ^{613}C and ^{615}N values (human and fauna) in the main text of manuscript.

We have added two new plots (Fig. 4A-B) to the manuscript displaying all the collagen and enamel results.

Line 133 of the text also mentions ^{618}O isotopic analysis was undertaken, however the results of this data are not discussed anywhere in the manuscript text. This analysis is also mentioned in section 3 of the Supplementary Information, but it is not clear exactly which individuals this was undertaken on, and no interpretation of the data generated is provided (as is done with the ^{613}C and ^{615}N data). Inclusion of this information is therefore needed.

We have now added the following to the results (lines 285-297):

*“Oxygen (^{618}O) values for humans ($n = 12$) and faunal ($n = 9$) tooth enamel for all three sites are variable, ranging from -3.7% to 0.9% and -7.8% to 2.2% respectively (Fig. 4B, Supplementary Dataset 13). Of the sites sampled, Molo Cave has the largest faunal isotopic dataset ($n = 6$) including wild taxa (i.e. *Dendrohyrax*, *Heterohyrax*) and domesticates (*Capra* and *Bos*). Caprine ^{618}O (-1.5% and 1.7%) are higher than that for the single *Bos* (-4.0%)*

sampled from the site. A similar distinction between caprines and cattle has been observed for Pastoral Neolithic sites in Kenya⁵⁷ and likely reflects differences in drinking behaviours, the proportions of ingested surface water and plant water, and/or taxonomic differences in fractionation. Given the known multifarious dietary and environmental influences on $\delta^{18}\text{O}$ and small number of enamel samples in this study, we cannot further assess these values, nor those obtained for Lukenya Hill and Cole's Burial, with great confidence. We present them for completeness and future use by other scholars undertaking more detailed $\delta^{18}\text{O}$ analysis of the assemblages, or compiling regional datasets."

4. Milk processing

With regards to potential consumption of different dairy products, the Discussion section of the paper mentions previous organic residue analysis undertaken, and the recovery of dairy lipids – do any of these studies show the presence thermally modified compounds, such as long-chain ketones? If so, this would potentially indicate processed milk products were being produced.

We thank the Reviewer for this raising this interesting point, to our knowledge the current published organic residue studies for Africa (Dunne et al. 2012, 2018; Grillo et al. 2020) do not discuss whether they detected thermally modified compounds.

It is also not clear from the discussion (paragraph starting line 317) whether the previous organic residue analysis work was undertaken on material from the same sites which were sampled for calculus here – or indeed if the studies were focused on East African material.

There is no overlap between the sites in this study and published lipid residue results for Africa.

Lines 330-337 now read:

"Combined with proteomic data from Sudan, and milk lipid residues from one sherd at Dongodien (GaJi4) in northern Kenya by 5000 cal. BP and three sherds at Ngamuriak (GuJf6) and Luxmanda in SW Kenya/northern Tanzania after c. 3000 cal. BP^{38,40}, these data point to milk consumption having been a widespread and persistent component of early herder lifeways. Together, these studies provide strong, multi-proxy evidence that people were regularly relying on access to animal milk throughout the expansion of pastoralism from the Sahara through eastern Africa with no detectable regional or temporal gaps."

We have also added details about the recent study by Grillo et al. 2020 (published after submission of this paper) which provides new evidence for the storage of dairy products in ceramics from pastoral sites in Kenya.

Lines 122-127: *"A recent lipid residue study of ceramics (n = 40) from early herder sites in the Lake Turkana Basin identified one sample with a $\Delta^{13}\text{C}$ value consistent with ruminant milk fats, suggesting some use of dairy products by 5000 BP in eastern Africa⁴⁰. The lipid analysis also returned three other positive results (and one possible positive result) for milk fats from c. 3000 year-old ceramics (n = 85) from southern Kenya and northern Tanzania⁴⁰."*

Minor comments:

- In the caption of Figure 2 it would be good to include a note that the pie charts represent the number of individuals with the presence of milk proteins.

The Figure caption now reads: “*Map of sites with calculus containing milk proteins. (A) Area of study in relation to the spread of cattle-based pastoralism across Africa (after²⁶). (B) Pie charts showing the number of individuals per site with milk proteins (shaded) proportionate to the total number of individuals that passed screening with Oral Signature Screening Database (see Methods and SI appendix for full details).*”

- Line 192: need to specify here which milk proteins were detected

Lines 211-12 now reads: “*We analysed dental calculus from individuals from seven sites across Sudan, and detected milk peptides (BLG and caseins) at the sites of Kadruka and Berber Meroitic Cemetery.*”

- Line 231: should read “BLG peptides” not just “BLG”
Corrected.

- Line 272: These references have not been included correctly and are missing from the reference list at the end of the manuscript
Corrected.

- Line 412: “stored -20 freezer” should read as “stored in a -20°C freezer”
Corrected.

- Supplementary Information, page 5, paragraph 3: “aparagene” should read “asparagine”
Corrected.

- Throughout the text (both the manuscript and the supplementary information) there appears to be a mix of UK English and American English spellings (e.g. demineralised and demineralized are both used, as are analysed and analyzed)
Corrected

Reviewers' Comments:

Reviewer #2:

Remarks to the Author:

The authors addressed convincingly all my previous comments. Just a few very minor edit suggestions:

Line 150:

Modify: "We used liquid chromatography mass spectrometry" to:

"We used liquid chromatography tandem mass spectrometry"

Line 160:

Modify: "including 6 that yielded milk proteins" to:

"including six whose dental calculus yielded milk proteins"

Line 163:

Modify: " Datasets 12-13) Stable oxygen ($\delta^{18}O$)" to:

" Datasets 12-13). Stable oxygen ($\delta^{18}O$)"

Line 171:

Modify: "Filled circles a represent sites" to:

"Filled circles represent sites"

REVIEWER COMMENTS

Reviewer #2 (Remarks to the Author):

The authors addressed convincingly all my previous comments. Just a few very minor edit suggestions:

Line 150:

Modify: "We used liquid chromatography mass spectrometry" to:
"We used liquid chromatography tandem mass spectrometry"
Corrected

Line 160:

Modify: "including 6 that yielded milk proteins" to:
"including six whose dental calculus yielded milk proteins"
Corrected

Line 163:

Modify: " Datasets 12-13) Stable oxygen (618O)" to:
" Datasets 12-13). Stable oxygen (618O)"
Corrected

Line 171:

Modify: "Filled circles a represent sites" to:
"Filled circles represent sites"
Corrected